# Hox genes control vertebrate body elongation by collinear Wnt repression

**Nicolas Denans[1,2†], Tadahiro Iimura[2,3], Olivier Pourquié[1,2,4,5,6,7]\***

[1]Institut de Génétique et de Biologie Moléculaire et Cellulaire, University of Strasbourg, Illkirch, France; [2]Stowers Institute for Medical Research, Kansas City, United States; [3]Division of Bio-Imaging Proteo-Science Center, Ehime University, Ehime, Japan; [4]Howard Hughes Medical Institute, Kansas City, United States; [5]Department of Anatomy and Cell Biology, University of Kansas Medical Center, Kansas City, United States; [6]Department of Pathology, Brigham and Woman's Hospital, Boston, United States; [7]Department of Genetics, Harvard Medical School, Boston, United States

**\*For correspondence:**
pourquie@genetics.med.harvard.edu

**Present address:** [†]Department of Developmental Biology, Stanford School of Medicine, Stanford, United States

**Competing interests:** The authors declare that no competing interests exist.

**Abstract** In vertebrates, the total number of vertebrae is precisely defined. Vertebrae derive from embryonic somites that are continuously produced posteriorly from the presomitic mesoderm (PSM) during body formation. We show that in the chicken embryo, activation of posterior *Hox* genes (paralogs 9–13) in the tail-bud correlates with the slowing down of axis elongation. Our data indicate that a subset of progressively more posterior *Hox* genes, which are collinearly activated in vertebral precursors, repress Wnt activity with increasing strength. This leads to a graded repression of the *Brachyury/T* transcription factor, reducing mesoderm ingression and slowing down the elongation process. Due to the continuation of somite formation, this mechanism leads to the progressive reduction of PSM size. This ultimately brings the retinoic acid (RA)-producing segmented region in close vicinity to the tail bud, potentially accounting for the termination of segmentation and axis elongation.

## Introduction

Body skeletal muscles and vertebrae form from a transient embryonic tissue called paraxial mesoderm (PM). The PM becomes segmented into epithelial structures called somites, which are sequentially produced in a rhythmic fashion from the presomitic mesoderm (PSM). The PSM is formed caudally during gastrulation by ingression of the PM progenitors located initially in the anterior part of the primitive streak (PS) and later, in the tail-bud (*Bénazéraf and Pourquié, 2013*). At the end of somitogenesis, the embryonic axis is segmented into a fixed species-specific number of somites which varies tremendously between species ranging from as little as ~32 in zebrafish to more than 300 in some snakes. The somites subsequently differentiate into their final vertebral and muscular derivatives to establish the various characteristic anatomical regions of the body. *Hox* genes code for a family of transcription factors involved in specification of regional identity along the body axis (*Mallo et al., 2012*; *Noordermeer and Duboule, 2013*). In mouse and chicken, the 39 *Hox* genes are organized in four clusters containing up to thirteen paralogous genes each. *Hox* genes exhibit both spatial and temporal collinearity, meaning that they are activated in a sequence reflecting their position along the chromosome and become expressed in domains whose anterior boundaries along the body axis also reflect their position in the clusters.

**eLife digest** In humans and other vertebrates, the number of bones (vertebrae) in the spine is determined early in development. The vertebrae form from blocks of tissue called somites that make segments along the body axis—a virtual line running from the head to the tail-end—of the embryo. The somites form as the embryo increases in length, with new somites forming periodically at the back near the embryo's tail-end.

A family of genes called the *Hox* genes are involved in controlling the formation of the somites. However, it is not known whether they directly control the number of somites that form, or whether they control the length of the body of the embryo.

Denans et al. studied the *Hox* genes in chicken embryos. The experiments suggest that the activation of some of the *Hox* genes in a structure called the tail-bud, which is found at the tail-end of the embryo, slow down the elongation of the body. The *Hox* genes achieve this by repressing the activity of a signaling pathway called Wnt so that Wnt activity in the tail-bud progressively decreases as the embryo develops.

The elongation of the body stops when the levels of a molecule called retinoic acid increase in the tail-bud, which causes the loss of the stem cells that are needed to make the somites. Denans et al.'s findings suggest that *Hox* genes influence the timing of the halt in elongation, which in turn is important for determining the total number of somites that form. Understanding how *Hox* genes control the formation of the cells that will make up the somites and influence Wnt signaling is a major challenge for the future.

Whether *Hox* genes control axis length and segment number has been controversial. Mouse mutants in which entire sets of *Hox* paralogs are inactivated show severe vertebral patterning defects but exhibit normal vertebral counts (*Wellik and Capecchi, 2003*; *McIntyre et al., 2007*). In contrast, precocious expression of *Hox13* genes in transgenic mice leads to axis truncation with reduced vertebral numbers (*Young et al., 2009*). Furthermore, mouse null mutations for *Hoxb13* or *Hoxc13* result in the production of supernumerary vertebrae (*Godwin and Capecchi, 1998*; *Economides et al., 2003*).

In chicken and fish embryos, the arrest of axis elongation has been linked to the inhibition of FGF and Wnt signaling in the tail-bud which leads to the down-regulation of the transcription factor *T/Brachyury* and of the Retinoic Acid (RA)-degrading enzyme *Cyp26A1* (*Young et al., 2009*; *Martin and Kimelman, 2010*; *Tenin et al., 2010*; *Olivera-Martinez et al., 2012*). Downregulation of *Cyp26A1* in the tail-bud ultimately leads to rising RA levels and to differentiation and death of the PM progenitors which terminates axis elongation. Premature exposure of the tail-bud to high RA levels in chicken or mouse embryos inhibits Wnt and FGF signaling and leads to axis truncation (*Tenin et al., 2010*; *Olivera-Martinez et al., 2012*; *Iulianella et al., 1999*) suggesting that the tail-bud must be protected from the differentiating action of RA. In the *Cyp26A1* null mutant mice, RA-signaling reaches the tail-bud, prematurely inducing the downregulation of FGF signaling and the increase of *Sox2* expression, resulting in axis truncation posterior to the thoracic level (*Abu-Abed et al., 2001*; *Sakai et al., 2001*). In chicken, the tail-bud starts to produce RA when explanted in culture after the 40-somite stage (*Tenin et al., 2010*). This late RA signaling activity in the tail-bud is involved in the termination of segmentation and axis elongation (*Tenin et al., 2010*; *Olivera-Martinez et al., 2012*). At the 40-somite stage, the mRNA for *Raldh2*, the RA-biosynthetic enzyme becomes expressed in the tail-bud potentially accounting for this late RA activity. What triggers this late expression of *Raldh2* in the chicken tail-bud is however unknown.

In vertebrates, the termination of axis elongation is accompanied by a progressive reduction in size of the PSM (*Gomez et al., 2008*; *Gomez and Pourquié, 2009*). The shrinking of the PSM which brings the segmented region producing RA in the vicinity of the tail-bud might also contribute to the raise in RA levels in the tail-bud and possibly to the late *Raldh2* activation in the tail-bud. Thus in the chicken embryo, the timing of elongation arrest (and hence the total number of somites formed) could be in part controlled by the kinetics of PSM shrinking. PSM size depends on the velocity of somite formation which removes cells anteriorly and on the flux of cells from the primitive streak and tail-bud generated during elongation which injects cells posteriorly. How this flux of progenitors ingressing in the PSM is

regulated over time, and which genes are regulating this process remain poorly understood. *Hoxb1-9* genes have been proposed to control cell ingression of paraxial mesoderm precursors from the epiblast during gastrulation (*Iimura and Pourquié, 2006*). However, *Hoxb1-9* genes are only expressed in anterior regions of the embryo precluding their playing a role in the control of axis termination.

Here, we first investigate the relationship between the speed of somite formation and of axis elongation. We show that, in the chicken embryo, activation of Abdominal B-like posterior *Hox* genes in the tail-bud correlates with the slowing down of axis elongation, while the speed of somitogenesis remains approximately constant. Our data indicate that a subset of progressively more posterior *Hox* genes, which are collinearly activated in vertebral precursors, repress Wnt and FGF activity with increasing strength, leading to a graded repression of the *Brachyury/T* transcription factor. This progressively reduces mesoderm ingression and cell motility in the PSM, thus slowing down the elongation process. Due to the continuation of somite formation at a steady pace, this mechanism leads to the progressive reduction of PSM size.

## Results

### Activation of posterior *Hox* genes correlates with axis elongation slowing down

We measured the variations of velocities of axis elongation and somite formation in time-lapse videos of developing chicken embryos during the formation of the first 30 somites (*Video 1*). The velocity of somite formation shows limited variation during this developmental window (*Tenin et al., 2010*) (*Figure 1A*, n = 4 embryos for each condition). In contrast, axis elongation velocity increases during the formation of the first 10 somites and then it decreases until the 25-somite stage, when it drops abruptly (*Figure 1A* and *Video 2*, n = 41 embryos). The number of PSM cells decreases over time (*Figure 1B*, n = 5 embryos for each condition) while no significant difference in cell proliferation or apoptosis in the PSM and tail-bud is observed (*Figure 1C–F*, n = 4 embryos for each condition). Cell motility, which has been implicated in the control of axis elongation (*Bénazéraf et al., 2010*), also decreased between 15 and 27 somites (*Figure 1G*, n = 4 embryos for each condition). Thus, a parallel decrease in cell motility and in cell flux to the PSM accompanies axis elongation slow down.

*Hoxb1-9* genes were shown to regulate cell flux to the PSM by controlling the timing of cell ingression from the epiblast (*Iimura and Pourquié, 2006*). Activation of *Hox* genes in the epiblast and tail-bud is collinear and occurs in two phases. First, the paralog groups 1 to 8 (and *Hoxb9*) are quickly activated within ten hours before the first somite formation (stage 7 HH [*Hamburger and Hamilton, 1992*]; *Figure 2*, n = 8 embryos for each condition). This phase is followed by a pause during formation of the first ten somites. Then between the 10 and 40-somite stage, the posterior *Hox* genes corresponding to the paralog groups 9–13 (and *Hoxc8* and *Hoxd8*) become subsequently activated in a slower phase which takes almost 48 hr (*Figure 2*, n = 8 embryos for each condition). *Hoxa13* is the first Hox13 activated at the 25-somite stage, when axis elongation slows down abruptly. Thus, there is a striking correlation between the timing of posterior *Hox* genes activation and the beginning of axis elongation slow down (*Figures 1A and 2*).

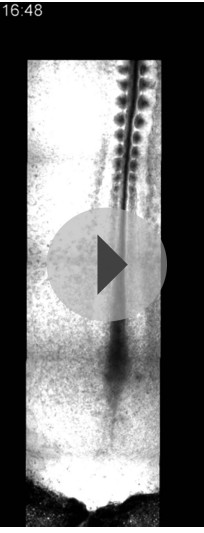

**Video 1.** Time-lapse video of an embryo from Stage 5 HH to 29 somites showing the different phases of axis elongation (Bright-field, ventral view, anterior is up). DOI: 10.7554/eLife.04379.003

### A subset of posterior *Hox* genes can regulate cell ingression and axis elongation

In order to test the role of posterior *Hox* genes on the control of cell ingression and cell motility in the developing chicken embryo, we used an in

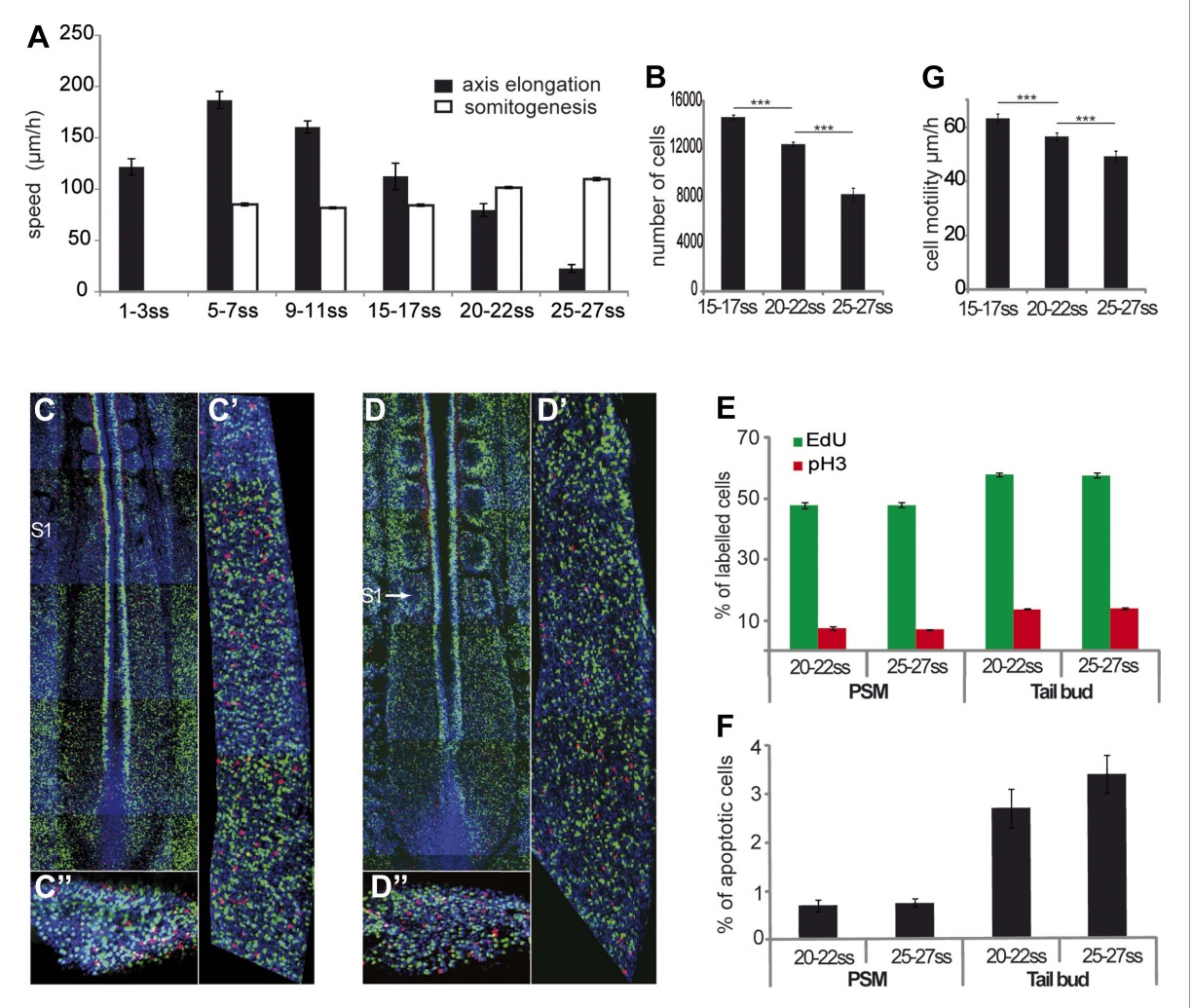

**Figure 1**. Slowing down of axis elongation correlates with decreasing cell ingression in the PSM. (**A**) Velocity of axis elongation and of somite formation. (**B**) PSM cell number. (**C**–**D**) Tiling of confocal sections of 20-somite (**C**) and 25-somite (**D**) stage embryos. EdU positive cells are labeled in green, phosphorylated histone H3 (pH3) in red, and nuclei in blue (DAPI). (**C'**, **D'**) Higher magnification of PSM regions used to quantify the proliferation. (**C''**, **D''**) Confocal sections of parasagittal cryosections of tail-bud used to quantify cell proliferation. (**E**–**F**) Quantification of cell proliferation (**E**) and apoptosis (**F**) in 20–22 and 25–27 somites chicken embryos. (**G**) Cell motility in the posterior PSM.

vivo electroporation technique, allowing to precisely target the paraxial mesoderm precursors in the epiblast of the anterior primitive streak (*Bénazéraf et al., 2010*) (*Video 3*). We developed a strategy allowing to overexpress two different sets of constructs in largely different populations of paraxial mesoderm cells by performing two consecutive electroporations of the paraxial mesoderm (PM) precursors of the epiblast of stage 4–5 HH embryos. Embryos are first electroporated on the left side of the primitive streak with a control Cherry construct, and then on the right side of the streak with a second vector expressing the yellow fluorescent protein Venus and a *Hox* construct (*Figure 3A*). This strategy results in essentially different PM cells expressing the two sets of constructs with the Cherry expressing cells enriched on the left side whereas Hox expressing cells are mostly found on the right side. When no *Hox* construct is present in the Venus vector, the Cherry and Venus-expressing populations of cells were observed to extend from the tail-bud to the same antero-posterior level of the paraxial mesoderm indicating that they began ingressing at the same time (*Figure 3B*, n = 8 embryos). In contrast, cells expressing Cherry were always extending more anteriorly than cells expressing Venus and one of the following posterior *Hox* gene: *Hoxa9, Hoxc9, Hoxd10, Hoxd11, Hoxc11, Hoxa13, Hoxb13,* or *Hoxc13*, indicating that these *Hox* genes can delay cell ingression of the

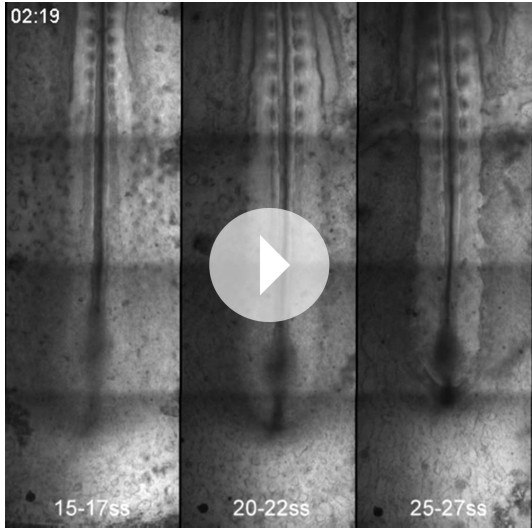

**Video 2.** Time-lapse videos showing axis elongation slow-down around the 25-somite stage. Bright-field imaging of chicken embryos at 15–17 somites (left panel), 20–22 somites (middle panel), and 25–27 somites (right panel) (ventral view, anterior is up).

PSM progenitors (*Figure 3B–C* n > 6 for each condition and not shown). This simply reflects the fact that cells ingressing later become located more posteriorly. Strikingly, the effect on ingression was progressively stronger when over-expressing more 5′ genes suggesting a collinear trend (*Figure 3C*). Inverting the order in which the constructs are electroporated did not affect the final phenotype. The distance between the anterior boundaries of the two domains was found to progressively increase with more posterior *Hox* genes as shown by measuring the ratio between Venus and Cherry posterior domains (*Figure 3A–C*). Over-expression of *Hoxa10*, *Hoxc10*, *Hoxa11*, *Hoxc12*, *Hoxd12* and *Hoxd13* showed no difference with the control Cherry vector (*Figure 3A–C*, n > 6 for each condition and data not shown). Using consecutive electroporation of *Hoxd10* and *Hoxc11* constructs labeled with Cherry and Venus, respectively, we observed that *Hoxc11* has a stronger effect on ingression than *Hoxd10* (*Figure 3—figure supplement 1*, n = 12 embryos). A similar result was observed when *Hoxa13* was compared to *Hoxc11* in the same assay (*Figure 3—figure supplement 1*, n = 6 embryos). Thus, a subset of posterior *Hox* genes is able to delay PSM cell ingression in a collinear manner.

To analyze the effect of posterior *Hox* genes on ingression, PM precursors were electroporated with Venus and a *Hoxa13* or a control construct and harvested after 5 hr when the electroporated cells start to ingress. No ectopic expression of laminin (*Figure 4A,B–C″*), acetylated tubulin (*Figure 4A, D–E″*), or E-cadherin (data not shown) was observed after *Hoxa13* over-expression. We compared the number of Venus-positive cells in epiblast vs primitive streak and mesoderm in embryo sections. The majority of *Hoxa13*-expressing cells were still found in the epiblast while control cells have ingressed into the primitive streak and mesoderm indicating that *Hoxa13* delays ingression by retaining cells in the epiblast (*Figure 4F–H*, n = 4 embryos for each condition). No up-regulation of the neural marker *Sox2* was observed in the tail-bud of embryos electroporated with *Hoxa13* (*Figure 4I–J*, n = 8 embryos for each condition) and very few cells were observed in the neural tube of embryos electroporated with Hox constructs (see *Figure 3B*, *Figure 4I–J*, Figure 6A–B, Figure 7D–J and Figure 9A and Videos 4–8). This indicates that the effect on ingression is not caused by the recruitment of PM precursor cells to a neural fate. Ingression of cells from the epiblast to the primitive streak occurs via an epithelium to mesenchyme transition which involves first destabilization and then complete loss of basal microtubules in these cells. This process has been shown to be regulated by a basally localized activity of the small GTPase RhoA (*Nakaya et al., 2008*). In order to test if the effect of the posterior *Hox* genes on delaying PSM progenitors ingression could involve regulation of microtubule stability, we used a dominant negative form of *RhoA* (DN-RhoA) as a tool to destabilize basal microtubules in the epiblast (*Nakaya et al., 2008*). We performed consecutive electroporations at stage 5 HH to overexpress a control Cherry vector in one population of PSM progenitors and *Hoxa13* with *DN-RhoA* vectors in another population and allowed the embryos to develop for 20 hr. We observed that the two populations of cells reach the same anterior level (*Figure 4K*, n = 5/5 embryos) indicating that these cells ingressed at the same time. Altogether, these results suggest that *Hox* genes control PSM progenitors ingression through the regulation of basal microtubule stability in the epiblast.

We next tested the effect of over-expressing posterior *Hox* genes on axis elongation (*Figure 5A–C*, *Video 4*, n = 47 embryos). Over-expression of either *Hoxa9*, *Hoxc9*, *Hoxd10*, *Hoxd11*, *Hoxc11*, *Hoxa13*, *Hoxb13* or *Hoxc13* but not of *Hoxa10*, *Hoxc10*, *Hoxa11*, *Hoxc12*, *Hoxd12* and *Hoxd13* in PM precursors caused a significant decrease of elongation velocity (*Figure 5A–C* and not shown). The effect of *Hox* genes becomes progressively stronger for more posterior genes (*Figure 5C*

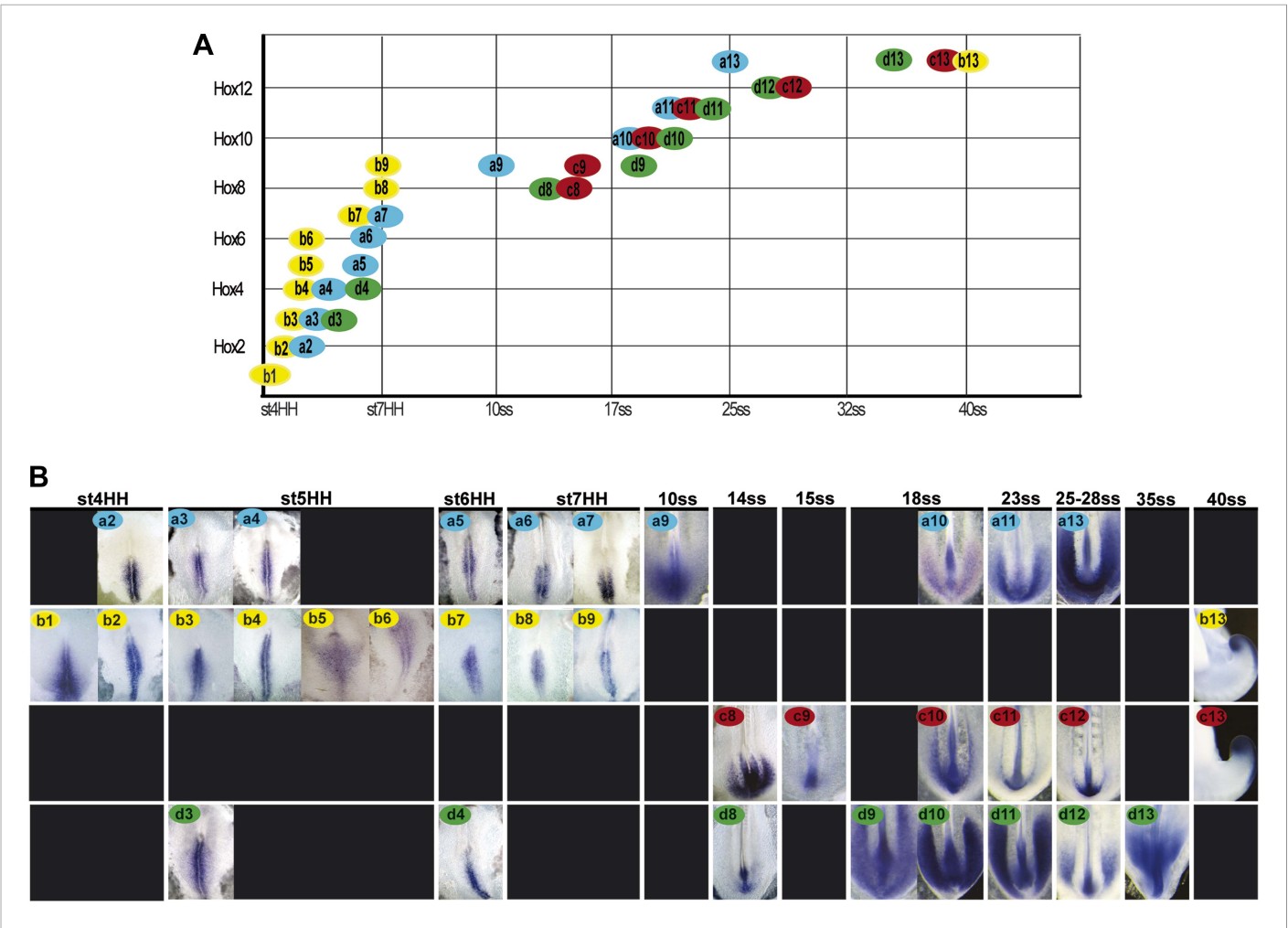

**Figure 2.** Collinear activation of *Hox* genes in paraxial mesoderm precursors. (**A**) Table showing the collinear onset of *Hox* genes expression in the epiblast/tail-bud generated from (**B**) Chicken embryos hybridized in whole-mount with *Hoxa* (blue), *Hoxb* (yellow), *Hoxc* (red), and *Hoxd* (green) probes. Each panel shows the beginning of activation of each *Hox* gene in paraxial mesoderm precursors in the epiblast of the anterior primitive streak or in the tail-bud. *Hox* probe used is indicated on the top of each panel. Anterior is up. Dorsal view.

and not shown, *Video 4*). Therefore, the same posterior *Hox* genes can alter cell ingression and axis elongation with a similar collinear trend (*Figures 3C and 5C*). The cell-autonomous control of ingression by posterior *Hox* genes (*Figure 4F–H*) is expected to reduce the supply of motile cells in the posterior PSM. This should slow down elongation movements and could explain why such a non-cell autonomous effect on axis elongation is observed while only 30–50% PM cells express the *Hox* constructs. These data suggest that a subset of posterior *Hox* genes controls the slowing down of axis elongation by regulating ingression of PM precursors.

In *Drosophila* and vertebrates, *Hox* genes expressed posteriorly can suppress the function of more anterior ones, a property termed phenotypic suppression or posterior prevalence (*Duboule and Morata, 1994*). We previously showed that posterior prevalence applies for the control of cell ingression by *Hoxb1-9* genes (*Iimura and Pourquié, 2006*). To test whether this property also applies to the posterior *Hox* genes with an effect on axis elongation, we performed consecutive electroporations first with a mix of *Hoxd10* and *Hoxc11* constructs (leading to expression in the same cells, in green *Figure 6A*) and then with a mix of *Hoxc11* and a control construct (a mutated *Hoxc11* unable to bind DNA (*Hoxc11mutH*), in red *Figure 6A*). We observed that cells over-expressing the two functional *Hox* genes reach the same anterior position as cells over-expressing *Hoxc11* and control (*Figure 6A,C*, n = 10 embryos). Thus *Hoxc11* function is dominant over *Hoxd10*.

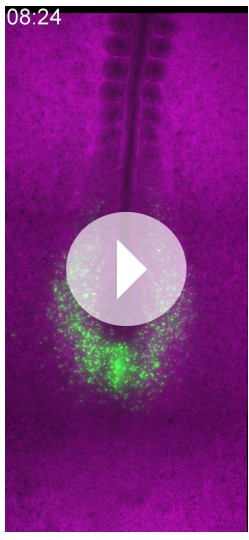

**Video 3.** Time-lapse video showing the precise targeting of PSM progenitors and the ingression of the epiblast cells to form the PSM. Bright-field (purple) merged with fluorescent images of PSM cell progenitors electroporated with a control *H2B-Venus* (ventral view, anterior is up) from stage 6 HH onwards.

Similarly, we observed dominance of *Hoxa13* over *Hoxc11* in the same assay (*Figure 6B,C*, n = 8 embryos). Therefore, posterior prevalence appears to generally apply for *Hox* control of cell ingression in the mesoderm (*Iimura and Pourquié, 2006*). As a result, the effect of *Hox* genes on cell retention in the epiblast should become progressively stronger as more posterior genes become activated.

## Pbx1 acts as a cofactor regulating cell ingression controlled by anterior *Hox* genes

Expression of anterior *Hox* genes in the primitive streak is maintained during the fast axis elonga-tion phase occurring during the formation of the first ten somites, suggesting that there must be a mechanism blocking their effect on ingression during this time window (*Figure 1A*). TALE (Three Amino-acid Loop Extension) family mem-bers have been shown to differentially interact with anterior and posterior *Hox* genes (*Chang et al., 1995*; *Moens and Selleri, 2006*). In chicken, the only TALE gene expressed in PM precursors is *Pbx1* which is detected in the primitive streak from stage 4 to 7 HH (*Figure 7A*, n = 8 embryos for each condition [*Coy and Borycki, 2010*]). Electroporation of a siRNA targeting *Pbx1* in the epiblast resulted in strong down-regulation of *Pbx1* (*Figure 7B–C*, n = 4 embryos for each condition). In consecutive electroporations performed first with Cherry and a control siRNA and then with Venus and a siRNA targeting *Pbx1*, cells electroporated with the *Pbx1* siRNA were found extending more anteriorly than control cells (*Figure 7D,K*, n = 19 embryos). The effect of *Pbx1* siRNA on ingression could be rescued by co-expressing *Pbx1* (*Figure 7E,K*, n = 16 embryos). We compared in consecutive electroporations the effect of expressing first a control siRNA with either *Hoxb7, Hoxb9, Hoxa9, Hoxc9, Hoxd10, Hoxd11, Hoxc11, Hoxa13, Hoxb13* or *Hoxc13*, and then the *Pbx1* siRNA with the same *Hox* gene. Cells co-expressing *Hoxb7* or *Hoxb9* and the *Pbx1* siRNA reached more anterior levels than cells co-expressing these *Hox* genes and the control siRNA (*Figure 7F–G, K*, n = 10 and 15 embryos respectively). In contrast, cells co-expressing either a control or the *Pbx1* siRNA together with a posterior *Hox* gene were found to extend up to the same anterior level (*Figure 7H–K* and not shown, n > 8 embryos for each condition). Over-expression of *Pbx1* in PM precursors after the 3-somite stage slowed down axis elongation (*Figure 7L* and *Video 5*, n = 12 embryos), suggesting that *Pbx1* can restore the effect of anterior *Hox* genes on ingression during this time window. Thus, *Hox*-dependent control of ingression in the paraxial mesoderm requires *Pbx1* for anterior but not for posterior *Hox* genes.

## *Hox* genes regulate axis elongation through collinear repression of Wnt/βcatenin

To identify effector targets regulated by posterior *Hox* genes, we electroporated epiblast PM progenitors in stage 5 HH embryos, either with a control *H2B-Venus* or with a *Hoxa13-IRES-H2B-Venus* vector, and we harvested embryos at 9 somites (*Figure 8A*). Venus-positive cells were sorted by Fluorescence Activated Cell Sorter (FACS) following tail dissociation and their transcriptome was analyzed using Affymetrix microarrays (*Figure 8A*, n = 2 × 2 arrays for each condition). The Wnt/βcatenin pathway targets, *Axin2, Fgf8*, and *Sp8* were down-regulated in *Hoxa13* over-expressing cells (*Table 1*, *Supplementary file 1*, *Figure 8B*) suggesting that posterior *Hox* genes might control axis elongation rate by progressively down-regulating the Wnt/βcatenin pathway. To test this hypothesis,

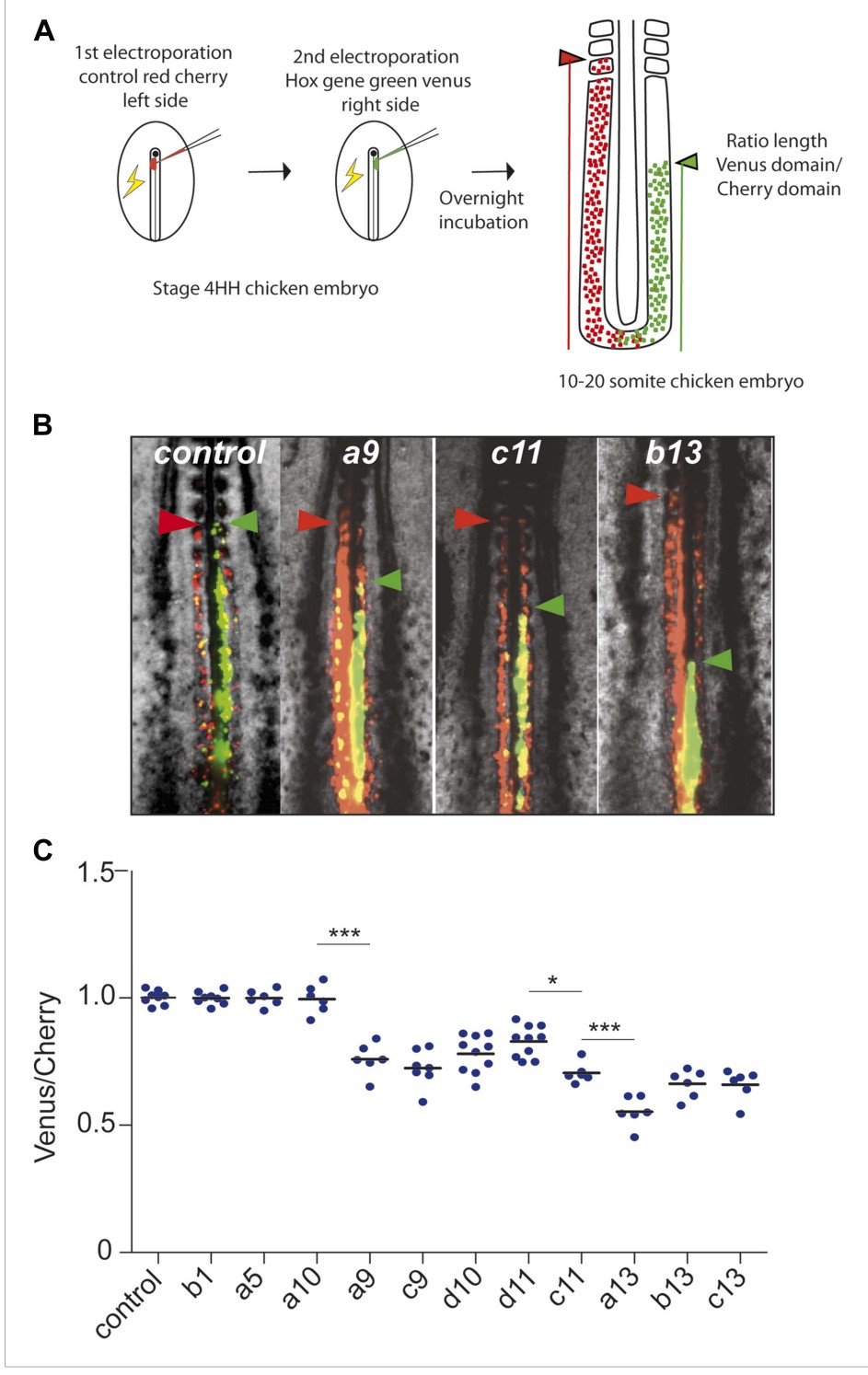

**Figure 3**. Posterior *Hox* genes can regulate cell ingression in a collinear fashion. (**A**) Consecutive electroporation protocol. The ratio of the green domain (green bar, *Hox* expressing) over the red domain (red bar, control vector) measures the ingression delay. (**B**) Embryos consecutively electroporated first with Cherry and then with Venus together with control, *Hoxa9*, *Hoxc11*, or *Hoxb13* vectors. Arrowheads: anterior boundary of Cherry (red) and Venus (green) domains. (**C**) Ratio of Venus over Cherry domains for posterior *Hox* genes. Dots: electroporated embryos. *Figure 3. continued on next page*

*Figure 3. Continued*

Bar indicates the mean. Stars: p-value of two-tailed Student's *t*-test applied between the different conditions.
*p < 0.05; **p < 0.01; ***p < 0.005. Error bars: standard error to the mean (SEM).
The following figure supplement is available for figure 3:

**Figure supplement 1**. The posterior *Hox* genes regulate cell ingression with increasing strength.

we first performed in situ hybridizations (ISH) for *Axin2, Fgf8,* and *T/Brachyury* that show that their expression in the tail-bud is down-regulated when *Hoxa13* becomes activated (*Figure 8—figure supplement 1A–D*, n = 8 embryos for each condition). Since the ISH technique is not quantitative enough to resolve slight differences, we performed quantitative Reverse Transcription PCR (qRT-PCR) on micro-dissected tail-buds from 10, 15, 20, and 25-somite stages for *Axin2, Fgf8,* and *T/Brachyury.* These experiments show a slight progressive down-regulation of these genes from the 10 to 20-somite stage followed by a significant decrease in gene expression at the 25-somite stage correlating with the slowing down of axis elongation as well as with the timing of posterior genes expression (*Figure 8C–E*, n = 5 embryos for each stage). Co-electroporation of *Hoxd10, Hoxc11,* or *Hoxa13* with *βcatLEF* (which activates the Wnt/βcatenin pathway [*Galceran et al., 2001*]) rescues axis elongation (*Figure 8F–H*, *Videos 6–8*, n = 41). We co-electroporated a Wnt/βcatenin firefly luciferase reporter (BATLuc) and a CMV-Renilla luciferase construct in PM progenitors together with either Venus or *Hoxa9, Hoxc9, Hoxd10, Hoxd11, Hoxc11, Hoxa13, Hoxb13,* or *Hoxc13*. These *Hox* genes induced a down-regulation of luciferase activity which increased in a collinear fashion (except for *Hoxd10* and *Hoxd11* which showed a weaker effect) (*Figure 8I* and *Figure 8—figure supplement 2A*, n = 83 embryos). All together, these results strongly suggest that the posterior *Hox* genes control axis elongation by modulating Wnt/βcatenin signaling activity. When co-expressing *Hoxa9* and *Hoxa13*, the Wnt-repressive effect was equivalent to that of *Hoxa13*, indicating that posterior prevalence also applies to Wnt repression (*Figure 6D–E*, n = 30 embryos). By expressing various amounts of *Hoxa13*, we observed that Wnt repression is independent of the quantity of protein expressed (*Figure 6G–H*, n = 62 embryos), suggesting that Hox proteins levels are saturating in our experiments. Therefore, the same posterior *Hox* genes can regulate ingression, axis elongation, and Wnt signaling with strikingly similar collinear trends.

We next analyzed how *Hox* genes interfere with Wnt function. *Hoxa13* ingression phenotype is rescued by an activated form of *Lrp6* or a stabilized form of *Ctbbn1* (*Figure 8J*, n = 42 embryos) but not by *Wnt3*a or *Wnt5a* (*Figure 8K*, n = 30 embryos). This suggests that, genetically, *Hox* genes act on Wnt signaling at the membrane level. Over-expression of the Wnt receptor *Fzd2* (down-regulated in *Hoxa13* over-expressing cells (*Figure 8B*, *Table 1* and *Supplementary file 1*)) with *Hoxa13* rescued Wnt repression (*Figure 8L*, n = 29 embryos). *Fzd2* is expressed in the tail-bud at 15 somites and down-regulated after 25 somites (*Figure 8—figure supplement 1E*, n = 8 embryos). Over-expression of the Wnt pathway component *Dact2* (which is expressed in the tail-bud from 25 somites onward and up-regulated in *Hoxa13* over-expressing cells [*Figure 8—figure supplement 1F*, n = 8 embryos, *Table 1*, *Supplementary file 1*]), repressed Wnt activity (*Figure 8M*, n = 9 embryos). In *Hoxa13* over-expressing cells, the FGF receptor *FGFR1*, its ligand *Fgf8*, and its targets *Etv1* and *Cyp26A1* as well as the FGF pathway component *Rasgrp3*, were down-regulated while the FGF/MAPK inhibitor, *Spred2*, was up-regulated (*Figure 8B*, *Table 1*, *Supplementary file 1*), indicating that *Hoxa13* can also inhibit FGF signaling. This inhibition is consistent with the down-regulation of PSM cell motility observed after *Hoxc11* or *Hoxa13* over-expression (*Figure 8—figure supplement 2B*, n = 20 embryos) (*Bénazéraf et al., 2010*). FGF down-regulation is expected since FGF and Wnt signaling reciprocally regulate each other in PM precursors (*Aulehla et al., 2003*; *Naiche et al., 2011*). Down-regulation of *Cyp26A1*, which degrades RA, can up-regulate RA signaling leading to repression of the Wnt pathway non cell-autonomously (*Iulianella et al., 1999*; *Young et al., 2009*; *Martin and Kimelman, 2010*). Together, these data suggest that posterior *Hox* genes act on a gene network converging toward autonomous and non-autonomous negative Wnt regulation.

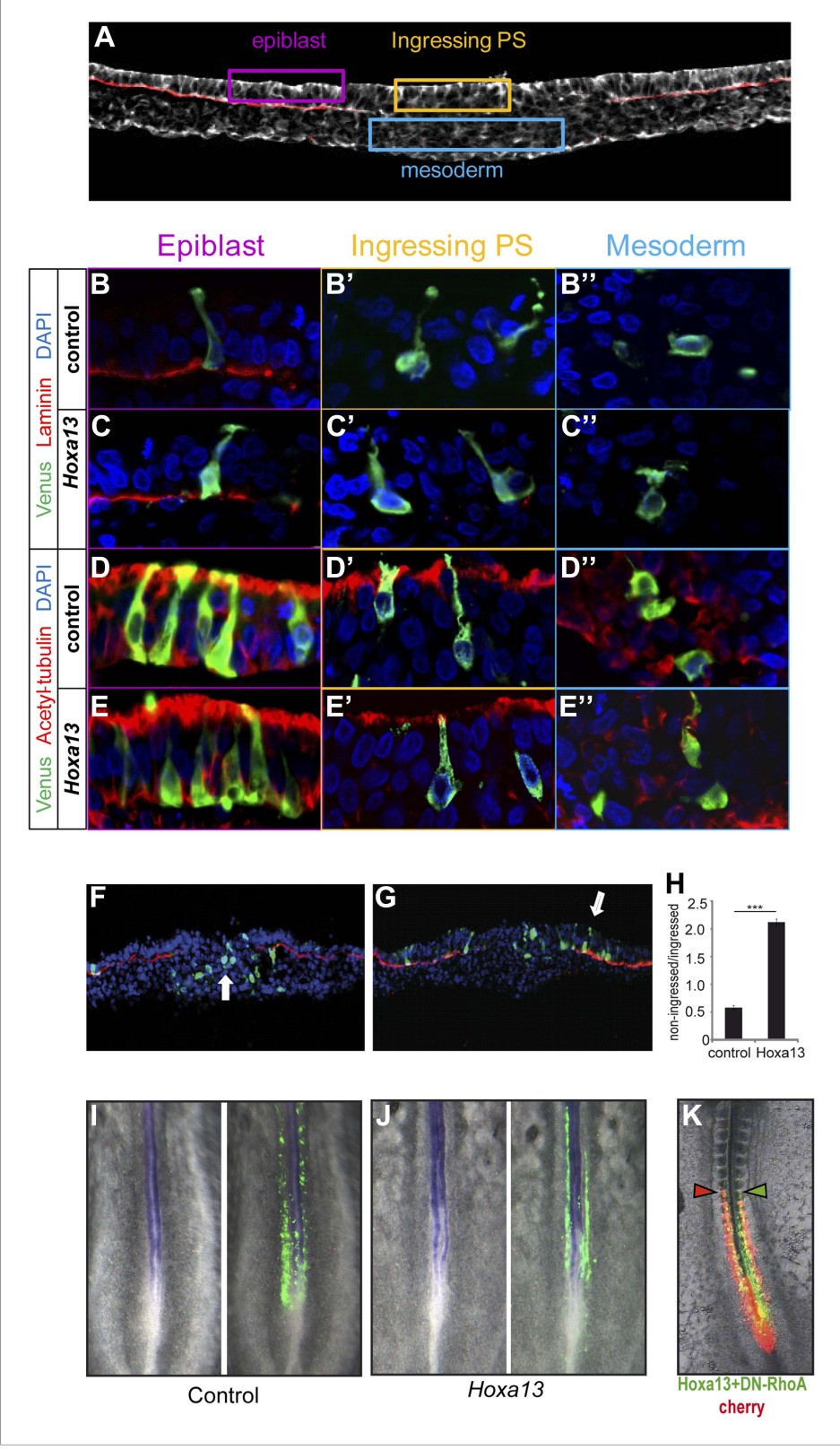

**Figure 4**. Epiblast cells overexpressing *Hox* genes do not convert to a neural fate. (**A**) Transverse section of a stage 7 HH chicken embryo labeled with phalloidin (white) to highlight the actin network and with laminin (red) to identify the epiblast basal membrane. Colored boxes indicate the different phases of differentiation of the mesoderm:
*Figure 4. continued on next page*

*Figure 4. Continued*

epiblast (purple), ingressing cells (yellow), and mesoderm (blue). (**B–E"**) Transverse sections at the PSM progenitors level 5 hr after electroporation of a control Venus or of *Hoxa13*. (**B-C"**) Laminin immunolabeling (red) after Venus (**B–B"**) or *Hoxa13* over-expression (**C–C"**). (**D–E"**) Acetylated α-tubulin immunolabeling (red) after Venus (**D–D"**) or *Hoxa13* (**E–E"**) over-expression. (**F–G**) Transverse cryosections of the anterior primitive streak of an embryo electroporated with Venus (**F**) or with Venus and *Hoxa13* (**G**). White arrow: cells ingressed in the primitive streak (**F**) and non-ingressed epiblast cells (**G**). Green: Venus; red: laminin; blue: nuclei. (**H**) Quantification of ingression in embryos electroporated with control or *Hoxa13*-expressing constructs. (**I–J**) In situ hybridization of 2-day old chicken embryos electroporated with Venus (**I**) or *Hoxa13*-Venus (**J**) expressing vectors. Left panel shows *Sox2* expression in the neural tube and tail-bud, and right panels show GFP immunohistochemistry. (**K**) Chicken embryo consecutively electroporated with a control (Cherry, red) and a mix of *Hoxa13*+*DN-RhoA* (Venus, green). Arrowheads: anterior boundary of Cherry (red) and Venus (green) domains. Stars: p-value of two-tailed Student's *t*-test applied between the different conditions. ***p < 0.005. Error bars: standard error to the mean (SEM).

## Gradual repression of *T/Brachyury* by posterior *Hox* genes regulates cell ingression and axis elongation

The T-box transcription factor *T* (aka *Brachyury*) is a well-characterized Wnt target which has been shown to control cell ingression to the mesoderm (*Wilson et al., 1995*; *Yamaguchi et al., 1999*). Q-PCR analysis of micro-dissected tail buds shows that T expression levels decrease between 10 and 20-somite stage and then significantly drop at the 25-somite stage (*Figure 8C*). Over-expressing *T* by electroporation often resulted in PM-expressing cells extending more anteriorly than control cells suggesting that they ingress earlier (*Figure 9A–B*, n = 6 embryos). Over-expression of *T* together with either *Hoxa9*, *Hoxd10*, *Hoxc11* or *Hoxa13* rescued the ingression delay (*Figure 9A–B*, n = 6, 11, 10 and 7 embryos respectively). *T* also rescued the elongation slow down observed after *Hoxa13* over-expression (*Video 6*, *Figure 9C*, n = 4 embryos). A lower dose of *T* (0.5 μg/μl) only led to partial rescue of the *Hoxa13* phenotype (*Figure 9C*, n = 4 embryos). Endogenous *T* expression is down-regulated in *Hoxa13* over-expressing cells FACS-sorted from electroporated embryos (*Figure 9D*, n = 2 FACS sorted cell samples for each condition). Over-expression of a reporter generated by fusing one kilobase of the chicken *T* promoter to the firefly luciferase (cTprLuc) together with *Hoxc11* or *Hoxa13* and the CMV-Renilla luciferase show *T* repression which is stronger for *Hoxa13* (*Figure 9E*, n = 19 embryos). Over-expression of *βcatLEF* leads to *T* up-regulation (*Figure 9F*, n = 20 embryos) and co-expression of *Hoxa13* with *βcatLEF* totally rescues *T* repression (*Figure 9F*, n = 20 embryos) suggesting that *Hox* genes down-regulate *T* expression by repressing the Wnt/βcatenin pathway. Over-expressing T had no effect on BATluc activation (*Figure 9—figure supplement 1*, n = 8 embryos). This argues that the effect of *Hox* genes on epiblast ingression involves quantitative regulation of *T* expression levels.

A *Hoxa13* truncated form is responsible for the dominant Hand-Foot-Genital syndrome in

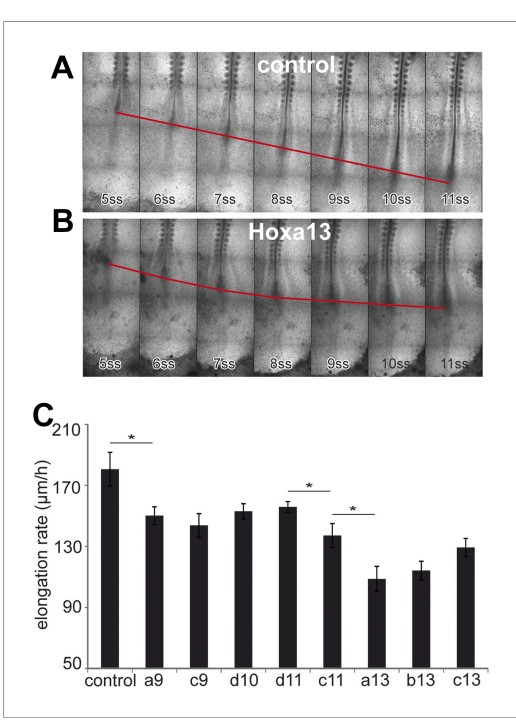

**Figure 5**. Posterior *Hox* genes control the axis elongation velocity in a collinear fashion. (**A–B**) Time-lapse series of chicken embryos electroporated either with control (**A**) or *Hoxa13* (**B**). Red line: position of Hensen's node. ss = somite-stage. (**C**) Velocity of axis elongation of embryos electroporated with either a control, *Hoxa9*, *Hoxc9*, *Hoxd10*, *Hoxd11*, *Hoxc11*, *Hoxa13*, *Hoxb13*, or *Hoxc13* expressing constructs. Stars: p-value of two-tailed Student's *t*-test applied between the different conditions. *p < 0.05. Error bars: standard error to the mean (SEM).

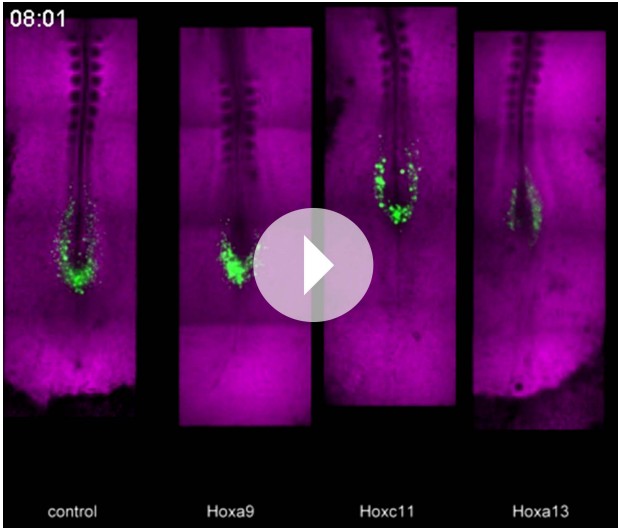

**Video 4.** Effect of Hoxa9, Hoxc11, and Hoxa13 electroporation on axis elongation and ingression. Bright-field (purple) merged with fluorescent images of PSM cell progenitors electroporated with either a control *H2B-Venus* (first panel from the left), *Hoxa9-ires2-H2B-Venus* (second panel from the left), *Hoxc11-ires2-H2B-Venus* (third panel) or a *Hoxa13-ires2-H2B-Venus* (right panel) constructs (green) (ventral view, anterior is up) from Stage 6 HH onwards. Over-expression of *Hoxa9, c11, and a13* affects ingression and axis elongation in a collinear fashion.

man (*Mortlock and Innis, 1997*). A similar truncation in the chicken homolog (*Hoxa13dn*) acts as a dominant-negative inhibiting the function of all *Hox13* genes (*de Santa Barbara and Roberts, 2002*). When over-expressed before activation of *Hox13* paralogs, *Hoxa13dn* had no effect on BATluc activity (*Figure 9G*, n = 18 embryos). However, co-expression with *Hoxa13* in similar conditions abolished Wnt repression (*Figure 9G*, n = 18 embryos). Similar truncations in chicken *Hoxd10* (*Hoxd10dn*) and *Hoxc11* ( *Hoxc11dn*) also exert a dominant-negative effect on their wild-type counterparts (*Figure 9H–I*, n = 38 embryos). We over-expressed *Hoxa13dn* alone or combined with either *Hoxc11dn* or with *Hoxc11dn* and *Hoxd10dn* along with BATLuc and CMV-Renilla constructs in PM precursors of the streak at stage 8 HH. Embryos were harvested at the 28-somite stage when most *Hox10-13* paralogs are expressed. Increasing the number of dominant-negative constructs results in a corresponding increase in luciferase activity (*Figure 9J*, n = 35 embryos). We next co-expressed the three dominant-negative vectors *Hoxd10dn, Hoxc11dn,* and *Hoxa13dn* together, along with Venus and FACS-sorted dissociated Venus-positive cells from tail-buds of 28-somite embryos. qRT-PCR analysis of *T, Axin2,* and *Fzd2* in the Venus-positive cells shows up-regulation of the three genes (*Figure 9K-L*, n = 4 embryos for each condition). All together these results argue that a subset of posterior *Hox* genes gradually represses Wnt/βcatenin signaling and consequently *T/Brachyury* in paraxial mesoderm precursors of the epiblast. This progressive repression leads to reduced cell ingression and cell motility in the PSM, resulting in a slowing down of axis elongation.

## The N-terminal region of posterior *Hox* genes but not the homeodomain is responsible for the repression of *T/Brachyury*

In order to identify the domain of posterior Hox proteins involved in repressing *T/Brachyury* expression, we generated chimera proteins where the different regions (N-terminal, homeodomain and C-terminal) of different posterior Hox proteins are swapped with the equivalent region of Hoxa5 which has no effect on axis elongation, Wnt activity and *T/Brachyury* expression (*Figure 10A–B*). Over-expression of cTprLuc along with a chimera where the homeodomain of Hoxa5 has been swapped with the one from Hoxa13 (Hoxa5Ha13) does not show any repression of luciferase activity while over-expression of a chimera where the homeodomain of Hoxa13 has been swapped with the one from Hoxa5 (Hoxa13Ha5) shows a strong repression of luciferase activity (*Figure 10A–B*, n = 35 embryos) suggesting that the homeodomain does not contain the major domain responsible for *T/Brachyury* repression. We next tested if either the N-terminal domain (N-ter) or the C-terminal domain (C-ter) is responsible for *T/Brachyury* repression. Overexpression of a chimera where the N-ter of Hoxa5 is swapped with the N-ter of Hoxa13 (NHoxa13HCa5) shows a strong repression of luciferase activity while a chimera where the C-ter of Hoxa5 is swapped with the C-ter of Hoxa13 (Hoxa5Ca13) does not show any repression (*Figure 10C–D*, n = 16 embryos) suggesting that the N-ter region of Hoxa13 contains the domain responsible for the repression of *T/Brachyury*. Sequence alignment of the N-terminal regions of Hoxa9, d10, c11, and a13 shows little conservation at the

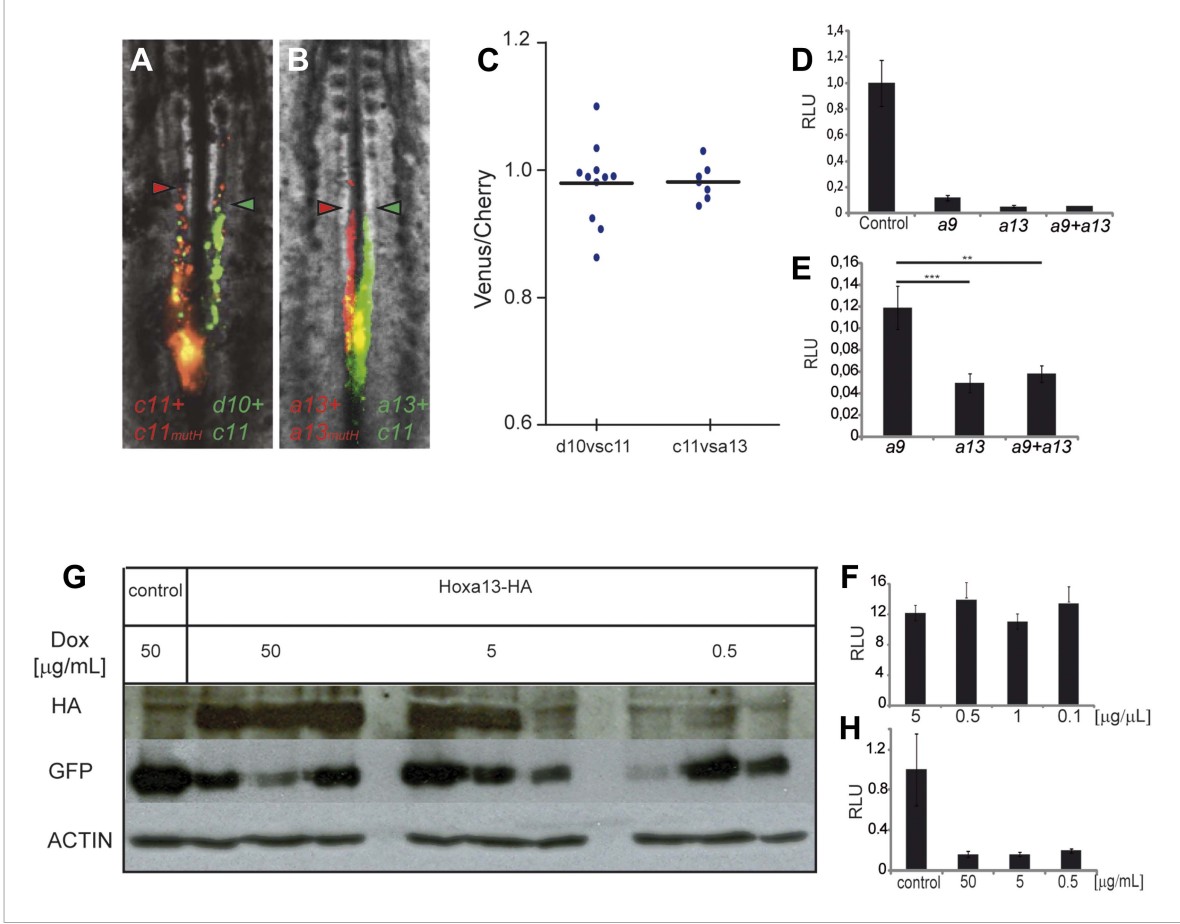

**Figure 6**. Posterior prevalence of posterior *Hox* genes. (**A**) Embryos consecutively electroporated first with *Hoxc11-Cherry + Hoxc11mutH-Cherry* and with *Hoxd10-Venus + Hoxc11-Venus* shown 24 hr after reincubation. (**B**) Embryos consecutively electroporated first with *Hoxa13-Cherry + Hoxa13mutH-Cherry* and then with *Hoxa13-Venus + Hoxc11-Venus* shown 24 hr after reincubation. Red arrowheads: anterior boundary of Cherry-expressing cells. Green arrowheads: anterior boundary of Venus-expressing cells. (**C**) Quantification of the ratio of Venus over Cherry expressing domains for the experiments shown in **A** and **B**. Each dot corresponds to one electroporated embryo and bar indicates the mean. (**D–E**) Luciferase assay measuring Wnt/βcatenin pathway activity after over-expression of the BATLuc construct together with a Renilla-expressing vector and either (**D**) control, *Hoxa9*, *Hoxa13* or the combination of *Hoxa9* and *Hoxa13* expressing vectors. (**E**) Blow-up of the samples shown in (**D**). (**F**) BATLuc assay with serial dilutions of the *Hoxa13* plasmid (in μg/μl on the x axis). (**G**) Western blot labeled with an anti-HA antibody showing embryos electroporated with *Hoxa13* under the control of a doxycycline-responsive promoter activated with different doses of doxycycline (in μg/ml). (**H**) BATLuc assay after *Hoxa13* over-expression under the control of a doxycycline-responsive promoter activated with different doses of doxycycline (in μg/ml on the x axis). Stars represent the p value of the two-tailed Student's *t*-test applied between the different conditions. **p < 0.01; ***p < 0.005. Error bars represent the standard error to the mean (SEM).

amino acid level suggesting that it is not a conserved amino acid domain but rather a structural domain that is responsible for the repression activity of these proteins (*Figure 10—figure supplement 1*). We next tested if the nature of the homeodomain could have a role in refining the level of repression of T by designing chimeras where the homeodomain of Hoxc11 and Hoxa13 was replaced by the homeodomain of Hoxa5 (Hoxc11Ha5 and Hoxa13Ha5, respectively) (*Figure 10E*). With the wild-type proteins, we observe a stronger downregulation of *T/Brachyury* with Hoxa13 than with Hoxc11 (*Figure 9E*). Surprisingly, when we overexpress Hoxc11Ha5 or Hoxa13Ha5 along with cTprLuc, we observe a stronger down-regulation of *T/Brachyury* with the chimera containing the Hoxc11 N-ter than with the one containing the Hoxa13 N-ter (*Figure 10E–F*, n = 26 embryos) suggesting that the homeodomain could be responsible for fine tuning *T/Brachyury* repression. Altogether our data suggest that the progressive deployment of posterior *Hox* genes in PM precursors during axis elongation leads to a collinear repression of the Wnt/βcatenin pathway and its target *T/Brachyury*.

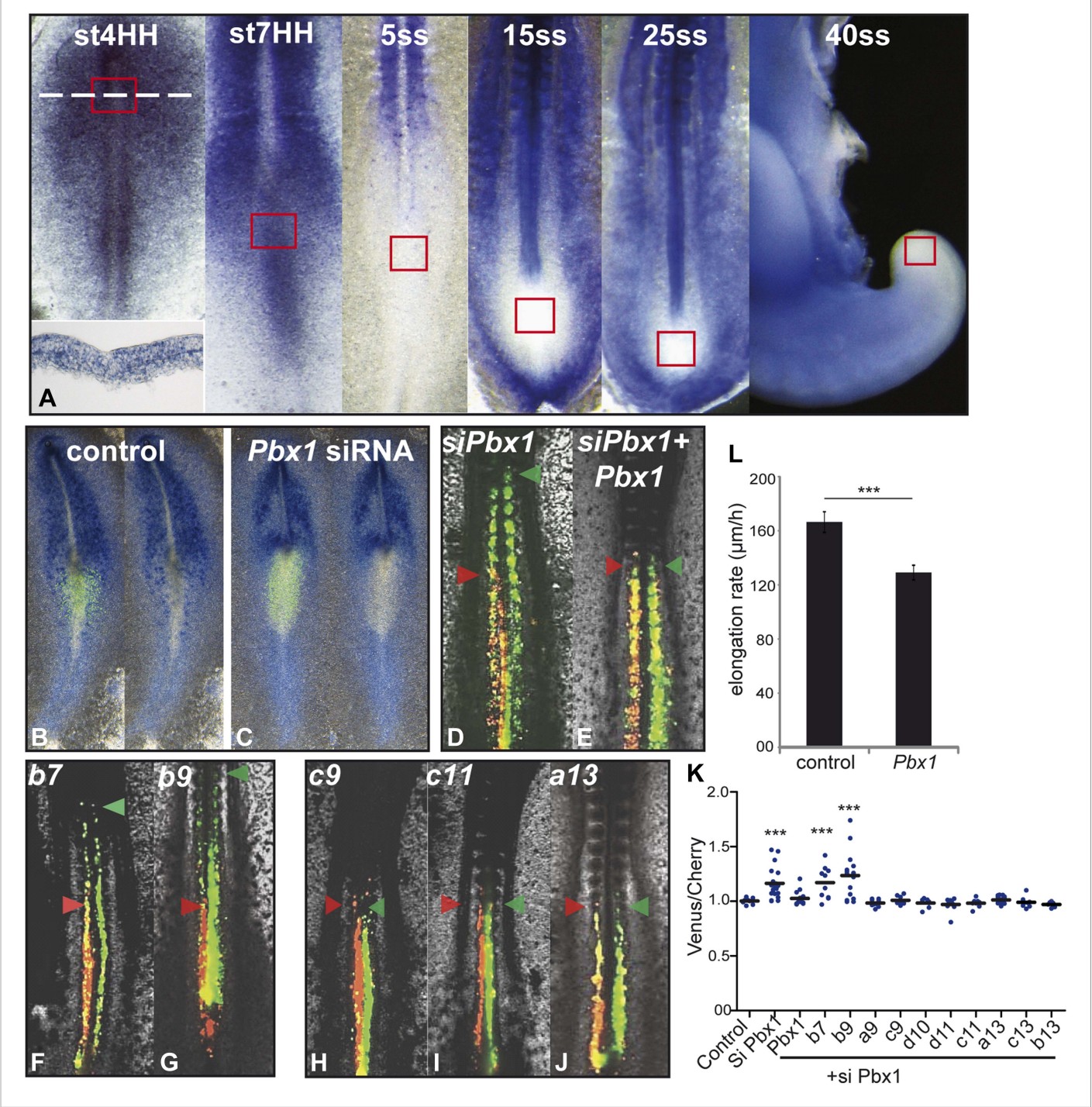

**Figure 7**. Control of ingression of PM precursors by anterior *Hox* genes is dependent on *Pbx1*. (**A**) *Pbx1* expression during somitogenesis. Red squares: region of PM progenitors. White dashed line: level of transverse section shown in bottom left. (**B–C**) *Pbx1* expression in stage 6–7 HH chicken embryos electroporated with Venus and control siRNA (**B**) or *Pbx1* siRNA (**C**). Left panels: Venus expression. (**D–J**) 2-day-old chicken embryos consecutively electroporated first with Cherry and a control siRNA and then with a *Pbx1* siRNA and a Venus construct either alone (**D**) or together with *Pbx1* (**E**), *Hoxb7* (**F**), *Hoxb9* (**G**), *Hoxc9* (**H**), *Hoxc11* (**I**), *Hoxa13* (**J**). Arrowheads: anterior boundary of Cherry (red) and of Venus (green) domains. (**K**) Ratio of Venus over Cherry expressing domains. Dots: electroporated embryos. Bar indicates mean. (**L**) Effect of *Pbx1* over-expression on axis elongation rate. Stars represent the p value of the two-tailed Student's *t*-test applied between the different conditions. ***p < 0.005. Error bars: standard error to the mean (SEM).

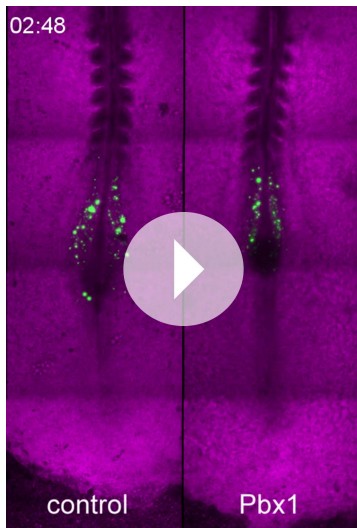

**Video 5.** Effect of Pbx1 over-expression between the 5- and 9-somite stage. Bright-field (purple) merged with fluorescent images of PSM cell progenitors electroporated with either a control *pBIC* (left panel) or a *Pbx1pBIC* (right panel) construct (green) (ventral view, anterior is up). Over-expression of *Pbx1* slows down axis elongation.

## Discussion

Here, we show that a subset of posterior *Hox* genes represses Wnt/T signaling with increasing strength showing a collinear trend. We observe a similar collinear effect of these posterior genes overexpressed in PM precursors on delaying their ingression in the PSM and on the slowing down of axis elongation. This inhibition of Wnt signaling is accompanied by a down-regulation of FGF signaling which was shown to control elongation velocity by regulating cell motility in the PSM (*Bénazéraf et al., 2010*). This suggests that posterior *Hox* genes are involved in the slowing down of axis elongation by acting both on the flux of cells in the posterior PSM and on the motility of PSM cells.

In the chicken embryo, PM precursors originate initially from the lateral epiblast which migrate toward the midline during formation of the primitive streak ( *Selleck and Stern, 1991*; *Hatada and Stern, 1994*). Around stage 4 HH, somite precursors begin to ingress from the superficial epiblast of the anterior primitive streak and posterior Node region (*Psychoyos and Stern, 1996*; *Iimura et al., 2007*). Two types of PM precursors have been identified in chicken and mouse embryos (*Wilson et al., 2009*). A first set derives from the Node/primitive streak border and exhibits long-term self-renewal properties (*Selleck and Stern, 1991*; *Cambray and Wilson, 2002*, *2007*). These cells express *Sox2* and *Brachyury* and they can contribute both to the PM (mostly to the medial part of the somites) and to the neural tube (*Ordahl, 1993*; *McGrew et al., 2008*; *Tzouanacou et al., 2009*; *Kondoh and Takemoto, 2012*; *Olivera-Martinez et al., 2012*). A second set derives from the anterior portion of the primitive streak and contributes to shorter clones restricted to the PM (*Iimura et al., 2007*; *McGrew et al., 2008*; *Tzouanacou et al., 2009*). After stage 4 HH, in the chicken embryo, the primitive streak begins to regress and after stage 13-14 HH, it becomes part of the tail-bud (*Schoenwolf, 1979*). At the 25-somite stage (stage 15 HH), the posterior neuropore closes and the tail-bud becomes enclosed into the tail fold. During these stages, PM precursors are continuously produced first by the primitive streak and then by the tail-bud. Fate mapping of the 25-somite stage tail-bud with quail-chick chimeras and diI labeling showed that the formation of the PM follows morphogenetic movements very similar to that seen earlier at the primitive streak level during gastrulation (*Catala et al., 1995*; *Knezevic et al., 1998*). After this stage, the remnant of the primitive streak becomes localized ventrally to form a structure known as the Ventral Ectodermal Ridge (VER) (*Schoenwolf, 1979*; *Goldman et al., 2000*).

Whether cell ingression continues after posterior neuropore closure to generate the PM is not well established. Knezevic et al. reported that cell ingression from the VER stops at stage 16 HH (26–28 somites) (*Knezevic et al., 1998*) but Ohta et al. demonstrated that ingression into the mesoderm continues in the VER up to the 40-somite stage (stage 20 HH) (*Ohta et al., 2007*). There is also some lineage continuity at the level of the PM precursors of the Node/primitive streak border which were shown to become internalized to become part of the chordo-neural hinge in the tail-bud. DiI labeling of the late chordo-neural hinge in stage 20–22 HH (40–45 somites) embryos showed that mesoderm cells are produced by this structure at late stages (*Olivera-Martinez et al., 2012*). The cellular organization of the chordo-neural hinge has not been characterized and whether mesoderm production by this structure occurs through ingression movements involving an epithelium to mesenchyme transition as is seen for the production of paraxial mesoderm from the primitive streak is not established. Overall, very little is known about the movements of cells in the tail-bud after the 25-somite stage in chicken and mouse embryos. The respective contribution of the VER and the CNH to the PM at these late stages has not been characterized.

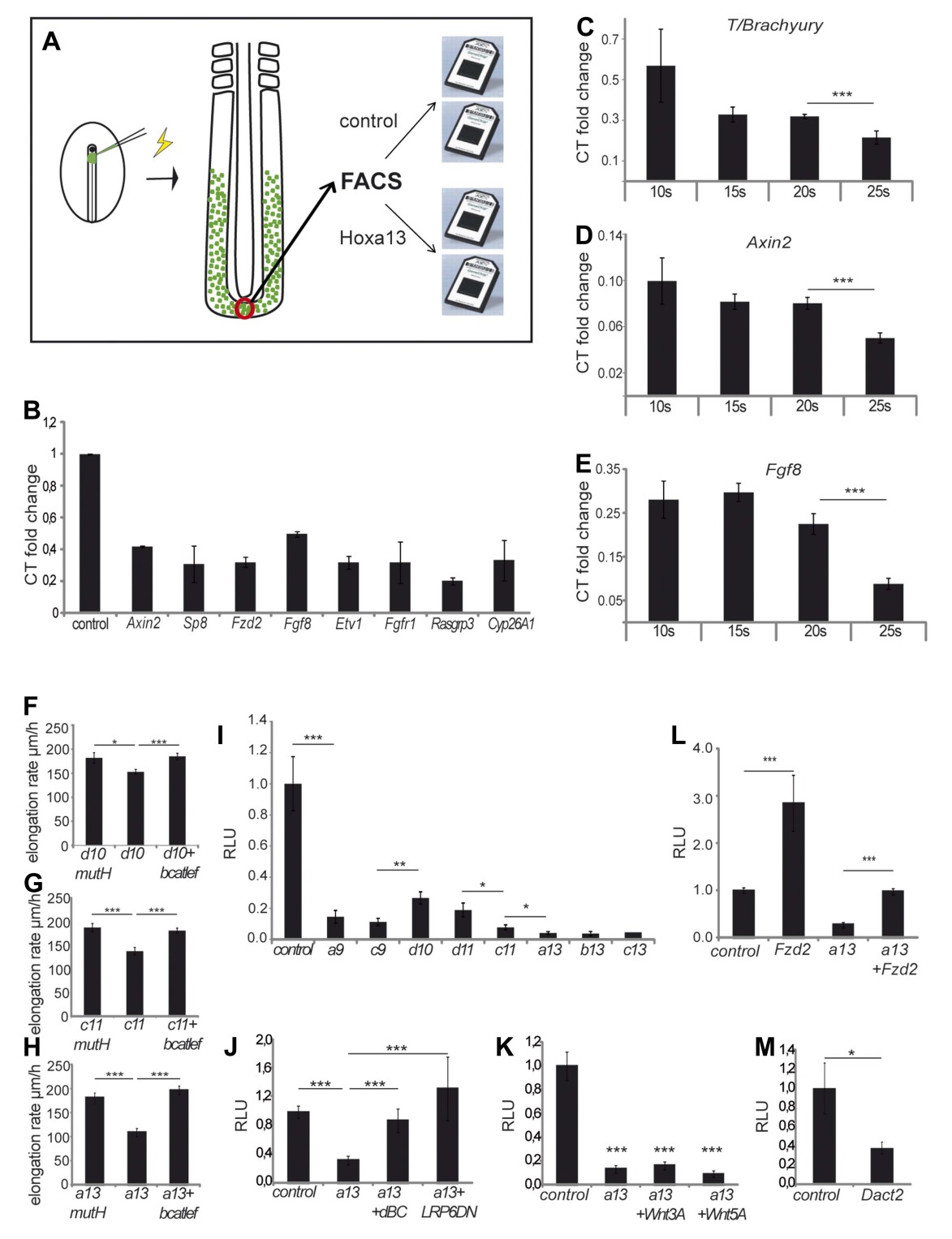

**Figure 8.** Collinear repression of Wnt/βcatenin signaling by posterior *Hox* genes. (**A**) Design of the microarray experiment. (**B**) Validation by Q-RT PCR of selected *Hoxa13* targets identified in the microarray experiment. (**C-E**) Q-RT PCR for (**C**) *T/Brachyruy*, (**D**) *Axin2*, and (**E**) *Fgf8* at 10, 15, 20, and 25-somite stage from microdissected tail-buds. (**F–H**) elongation velocity of embryos over-expressing (**F**) *Hoxd10mutH, Hoxd10* or *Hoxd10+βcatLEF*, (**G**) *Hoxc11mutH, Hoxc11* or *Hoxc11+βcatLEF*, (**H**) *Hoxa13mutH, Hoxa13* or *Hoxa13+βcatLEF*. (**I**) Luciferase assay measuring Wnt/βcatenin activity after over-expression of BATLuc together with CMV-Renilla and either control, *Hoxa9, Hoxc9, Hoxd10, Hoxd11, Hoxc11, Hoxa13, Hoxb13,* or *Hoxc13*. (**J–M**)
*Figure 8. continued on next page*

*Figure 8. Continued*

Luciferase assay measuring Wnt/βcatenin activity after over-expression of BATLuc and CMV-Renilla and control, *Hoxa13*, *Hoxa13+dBC*, or *Hoxa13+Lrp6ΔN* (**J**), or control, *Hoxa13*, *Hoxa13+Wnt3a* or *Hoxa13+Wnt5a* (**K**), or control, *Fzd2*, *Hoxa13*, or *Hoxa13+Fzd2* (**L**) or control and *Dact2* (**M**). Firefly luciferase intensity values have been normalized to their respective Renilla values (RLU). Controls have been set to 1. Stars: p value of the two-tailed Student's *t*-test applied between the different conditions. *p < 0.05; **p < 0.01; ***p < 0.005. Error bars represent standard error to the mean (SEM).
The following figure supplements are available for figure 8:

**Figure supplement 1**. The Wnt signaling is repressed when posterior *Hox* genes are activated.
**Figure supplement 2**. Collinear repression of Wnt signaling and cell motility by posterior *Hox* genes.

If Knezevic et al. are correct, it could be that the action of posterior Hox genes on ingression ends at the 25-somite stage when *Hoxa13* is first expressed. The strong effect of *Hoxa13* on ingression might trigger the arrest of cell ingression leading to the slowing down of axis elongation observed at this stage. The resulting imbalance between the velocity of somitogenesis and of axis elongation could account for the progressive shortening of the PSM observed during the production of the next 25–28 somites. Alternatively, it could be that, as suggested by Ohta et al. and by Olivera-Martinez et al., ingression continues up to the 40–45 somite stage. In this case, *Hoxa13* expression at the 25-somite stage would only significantly reduce the rate of cell ingression into the PM. At the 40–43 somite stage, *Hoxb13* and *Hoxc13* become expressed in the tail-bud potentially terminating further ingression. Whether Hox13 genes might regulate the late ingression of PM at the level of the VER, as shown by Ohta et al., or at the level of the CNH as proposed by Olivera-Martinez remains to be established. In both cases, however, the PSM is expected to shrink in response to Hox13 genes.

Even though our experiments do not directly address the process whereby axis elongation stops, they suggest that Wnt and FGF repression in the tail-bud, which signals termination of axis formation (**Olivera-Martinez et al., 2012**), could be mediated by posterior *Hox* genes. By reducing the flux of cells to the PSM and their motility, posterior *Hox* genes can indirectly control its progressive shortening. Furthermore, the inhibition of FGF and Wnt signaling which are required for the segmentation clock oscillations provides an explanation for the arrest of somite formation before the complete exhaustion of the PSM described in avians (**Bellairs, 1986**; **Tenin et al., 2010**). In vivo, the downregulation of the FGF target *Cyp26A1* downstream of Hox13 genes (this report, **Young et al., 2009**) would leave the tail-bud more vulnerable to the increase of RA. Whether, the raise in RA levels caused by bringing the segmented region closer is also responsible for *raldh2* activation in the late tail-bud remains to be explored. In such scenarios, posterior *Hox* genes indirectly control the termination of axis elongation and hence the segment number in the chicken embryo.

**Table 1**. List of selected genes of the Wnt and FGF pathways down-regulated or up-regulated following over-expression of *Hoxa13* in tail-bud cells

| Gene | Average (control) | Standard Dev (control) | Average (Hoxa13) | Standard dev (Hoxa13) | Fold change |
|------|-------------------|------------------------|------------------|------------------------|-------------|
| *Sp8* | 949.9 | 279.2 | 483.8 | 21.4 | 0.51 |
| *Fzd2* | 139.7 | 10.7 | 78.5 | 8.6 | 0.56 |
| *Axin2* | 857.8 | 42.5 | 677.0 | 99.1 | 0.79 |
| *Dact2* | 415.8 | 134.4 | 989.8 | 270.1 | 2.38 |
| *Cyp26a1* | 625.1 | 258 | 102.9 | 13 | 0.16 |
| *Fgf8* | 1523.9 | 159.3 | 591.2 | 65 | 0.39 |
| *Etv1* | 296.6 | 113.2 | 155.1 | 23.8 | 0.52 |
| *Fgfr1* | 145.9 | 5.8 | 80 | 0.6 | 0.55 |
| *Rasgrp3* | 1441.3 | 671.8 | 362.8 | 218.8 | 0.25 |

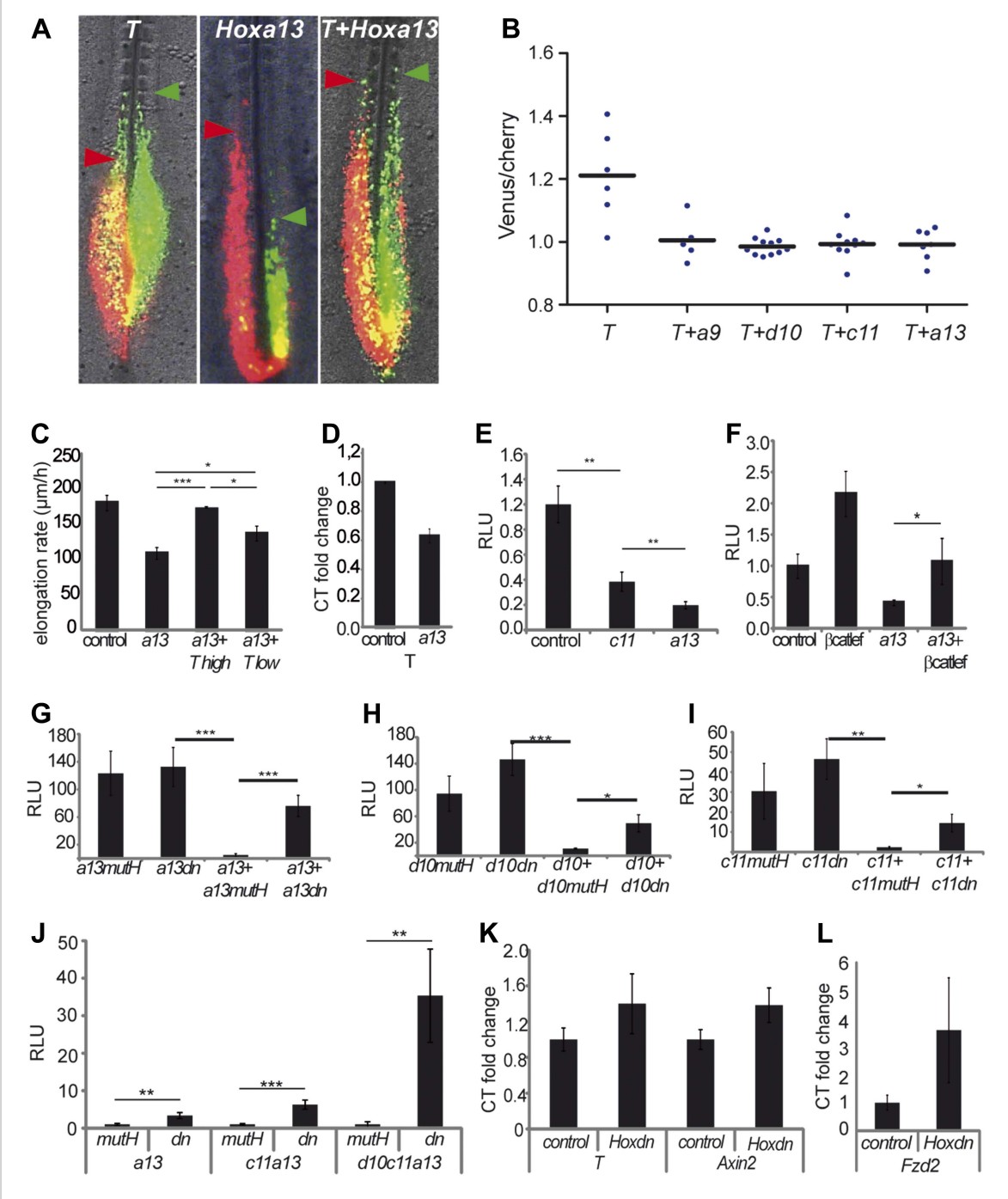

**Figure 9**. *Hox* genes effect on axis elongation involves *Brachyury* regulation downstream of the Wnt/βcatenin pathway. (**A**) Consecutive electroporation of PM precursors with Cherry and then with Venus together with *T* (left panel), *Hoxa13* (middle), or a combination of the two vectors (right). Arrowheads: anterior boundary of Cherry (red) and Venus (green) domains. (**B**) Ratio of Venus over Cherry domains. Dots: electroporated embryos. Bar indicates the mean. (**C**) Axis elongation velocity of embryos electroporated with control, *Hoxa13*, or co-electroporated with *Hoxa13* and either high or low dose of *T*. (**D**) Q-RT PCR quantification of *T* expression in control or *Hoxa13*-expressing PM progenitor cells. (**E–F**) Luciferase activity (RLU) after over-expression of cTprLuc and CMV-Renilla together with either (**E**) control, *Hoxc11* or *Hoxa13* or (**F**) control, *βcatLEF*, *Hoxa13* or *Hoxa13+βcatLEF*. (**G–I**) Luciferase assay measuring Wnt/βcatenin activity after over-expression of BATLuc and CMV-Renilla and (**G**) *Hoxa13mutH*, *Hoxa13dn*, *Hoxa13+Hoxa13mutH* or *Hoxa13+Hoxa13dn*. (**H**) *Hoxd10mutH*, *Hoxd10dn*, *Hoxd10+Hoxd10mutH* or *Hoxd10+Hoxd10dn*, (**I**) *Hoxc11mutH*, *Hoxc11dn*, *Hoxc11+Hoxc11mutH* or *Hoxc11+Hoxc11dn*. (**J**) Luciferase assay measuring Wnt/βcatenin activity from 28-somite stage dissected tail-buds after over-expression of BATLuc and CMV-Renilla constructs and either *Hoxa13mutH* or *Hoxa13dn*, or *Hoxa13mutH* with *Hoxc11mutH* or *Hoxa13dn* with *Hoxc11dn*, or *Hoxa13mutH* with

*Figure 9. continued on next page*

*Figure 9. Continued*

*Hoxc11mutH* and *Hoxd10mutH* or *Hox13dn* with *Hoxc11dn* and *Hoxd10dn*. (**K**, **L**) Q-RT PCR quantification of *T, Axin2* (**K**), and *Fzd2* (**L**) expression in PM progenitors co-expressing either *Hoxa13mutH* with *Hoxc11mutH* and *Hoxd10mutH* or *Hoxa13dn* with *Hoxc11dn* and *Hoxd10dn*. Stars: p-value of the two-tailed Student's *t*-test applied between the different conditions. *p < 0.05; **p < 0.01; ***p < 0.005. Error bars: standard error to the mean (SEM).

The following figure supplement is available for figure 9:

**Figure supplement 1**. Overexpression of T has no effect on Wnt activity.

In mouse embryos, over-expression of Hox13 genes results in axis truncation posterior to the thoracic level (*Young et al., 2009*). Remarkably, overexpression of *Hoxa13*, *b13*, and *c13* from the same promoter in transgenic mice results in truncations at different antero-posterior levels (*Young et al., 2009*), arguing for different truncation efficiency of the mouse Hox13 proteins. This is highly reminiscent of our observations showing different quantitative effects of the overexpression of the same three Hox13 genes in chicken embryos. Duplications and deletions of regions of the mouse *Hoxd* cluster lead to heterochronic expression of posterior *Hoxd* genes in the tail-bud yet they do not seem to alter segment numbers (*Spitz et al., 2001*; *Kmita et al., 2002*; *Tarchini and Duboule, 2006*; *Tschopp et al., 2009*). This is also consistent with our observations that *Hoxd* genes have limited effect on axis elongation in our experiments.

In transgenic mice overexpressing *Hoxc13*, Wnt targets and the FGF target *Cyp26A1* were also found to be down-regulated (*Young et al., 2009*) as observed in chicken embryos overexpressing *Hoxa13*. This argues for a conserved role of posterior Hox proteins in the repression of the Wnt and FGF pathway between chicken and mouse embryos. In mouse embryos however, no strong *raldh2* expression or late RA production is detected in the tail bud (*Tenin et al., 2010*) and axis elongation continues for a longer time resulting in tail formation. Moreover, *raldh2 −/−* mouse embryos which lack RA production during posterior body formation can form normal tails, suggesting that RA is not involved in axis termination in mouse (*Cunningham et al., 2011*). In mouse embryos, *Wilson and Beddington (1996)* initially reported an arrest of ingression when the posterior neuropore closes at the 30-somite stage (*Wilson and Beddington, 1996*), but *Cambray and Wilson, 2002* subsequently provided evidence for continued ingression of cells in the PM after this stage (*Cambray and Wilson,*

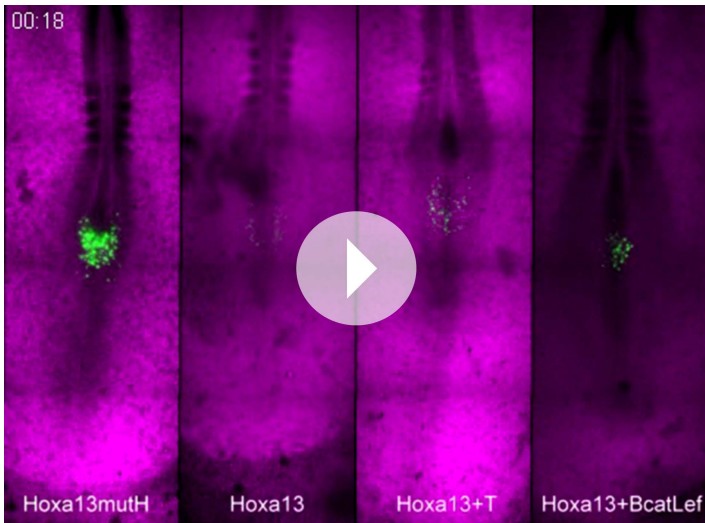

**Video 6.** Activation of Wnt/βcatenin signaling and T over-expression rescue Hoxa13 axis elongation phenotype. Bright-field (purple) merged with fluorescent images of PSM cell progenitors electroporated with *Hoxa13mutH-ires2-H2B-Venus* (left panel), *Hoxa13-ires2-H2B-Venus* (second panel), *T* and *Hoxa13-ires2-H2B-Venus* construct (third panel) or *βcatLEF* and *Hoxa13-ires2-H2B-Venus* construct (right panel) (green) (ventral view, anterior is up) from Stage 6 HH onwards.

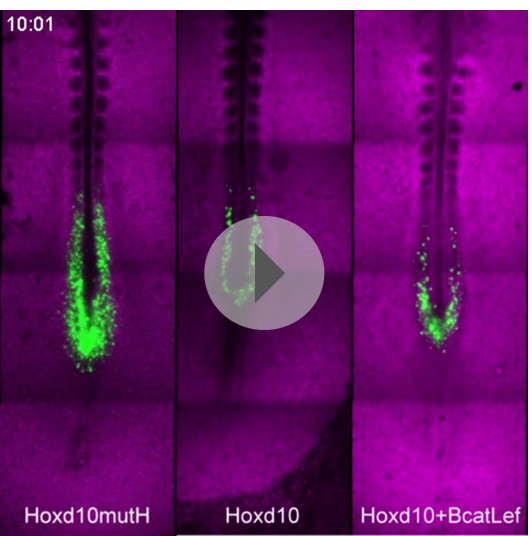

**Video 7.** Activation of the Wnt/βcatenin pathway rescues the axis elongation phenotype due to Hoxd10 over-expression. Bright-field (purple) merged with fluorescent images of PSM cell progenitors electroporated with *Hoxd10mutH-ires2-H2B-Venus* (left panel), *Hoxd10-ires2-H2B-Venus* (middle panel) or *βcatLEF* and *Hoxd10-ires2-H2B-Venus* construct (right panel) (green) (ventral view, anterior is up) from Stage 6 HH onwards.

*2002*). Thus, in mouse embryos, termination of axis elongation could simply result from exhaustion of PM progenitors caused by the slowing of axis elongation triggered by posterior *Hox* genes acting on cell ingression and motility. Remarkably, among amniotes, many species such as lizards, rodents, or monkeys bear a long tail whereas others such as birds or humans do not. Closely related species such as monkeys and apes can differ by the presence of a tail suggesting that the genetic switch involved in the control of tail formation is quite simple. Whether this switch involves an RA-dependent elongation arrest mechanism as seen in chicken and whether this control depends on posterior *Hox* genes is an attractive possibility which remains to be investigated.

Our work provides evidence for functional collinearity in the control of axis elongation by posterior *Hox* genes. Our data also suggest that our overexpression conditions are saturating (*Figure 6*), abolishing any effect of gene dosage of the overexpressed *Hox* genes. This confirms previous results published in *Iimura and Pourquié, 2006* showing that *Hoxb1-9* gene expression driven by promoters of different strength (CMV, TK, and CAGGS) leads to similar ingression phenotypes. Together, this suggests that the information driving the quantitative effects of Hox proteins on Wnt repression, cell ingression, and elongation is built in the structure of the proteins themselves rather than reflecting the actual amounts of Hox proteins present. This functional collinearity might be related to the recently described structural collinearity of binding specificities reported for fly Hox proteins (*Slattery et al., 2011*).

Our work suggests that low amounts of posterior Hox protein levels could be saturating in vivo. This is consistent with the analysis of paralog knock-out experiments showing that leaving only one single wild-type allele leads to a much milder phenotype than the deletion of an entire paralog group (*Wellik and Capecchi, 2003*; *McIntyre et al., 2007*). Also, increasing Hox doses by adding an extra mouse or human HoxD cluster does not alter the vertebral formula (*Spitz et al., 2001*; *Kmita et al., 2002*; *Tarchini and Duboule, 2006*; *Tschopp et al., 2009*). The fact that low levels of *Hox* proteins are saturating could confer great robustness to the system consistent with the extreme stability of intraspecific vertebral formula. That 8 of the 16 posterior *Hox* genes from all posterior paralog groups except Hox12 show an effect in the ingression, elongation, and Wnt signaling assays argue for an extreme redundancy of the system that could further explain the intraspecific robustness of the vertebral formula.

We observe a trend showing an increasing strength of the effects on cell ingression, Wnt repression, and axis elongation when overexpressing progressively more 5′ Hox genes. This increasing trend can be partly accounted for by the posterior prevalence of posterior *Hox* genes observed in the control of ingression and in the repression of Wnt signaling for genes of different paralog groups. However, different quantitative effects were observed for genes from the same paralog groups arguing against a simple posterior prevalence model. This is in line with the result of inactivation of the entire paralogs groups such as *Hox10* or *Hox11* which demonstrates specific properties of each of these paralog groups arguing against a simple posterior prevalence model functioning in vertebral patterning (*Wellik and Capecchi, 2003*; *McIntyre et al., 2007*).

We identify a role for the TALE protein Pbx1 in the control of cell ingression from the epiblast into the PSM by anterior but not posterior *Hox* genes. TALE homeoproteins have been shown to act as co-factors able to enhance DNA binding specificity of Hox genes (*Moens and Selleri, 2006*).

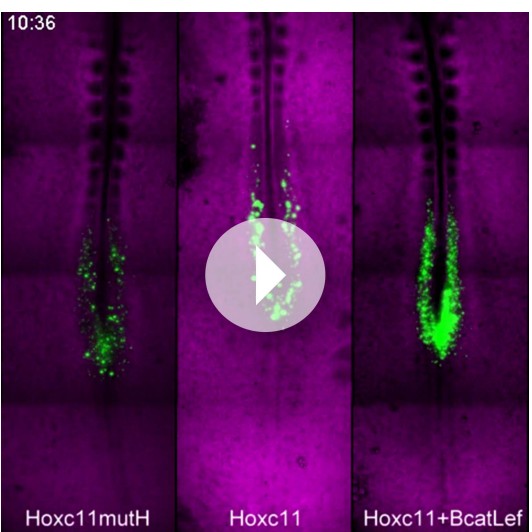

**Video 8.** Activation of the Wnt/βcatenin pathway rescues the axis elongation phenotype due to Hoxc11 over-expression. Brightfield (purple) merged with fluorescent images of PSM cell progenitors electroporated with *Hoxc11mutH-ires2-H2B-Venus* (left panel), *Hoxc11-ires2-H2B-Venus* (middle panel), or *βcatLEF* and *Hoxc11-ires2-H2B-Venus* construct (right panel) (green) (ventral view, anterior is up) from Stage 6 HH onwards.

Pbx proteins bind anterior Hox proteins via a specific hexapeptide sequence (*Chang et al., 1995*). The null mutation of *Pbx1* in mouse leads to patterning defects of the axial skeleton but axis length appears essentially normal (*Selleri et al., 2001*). In contrast, double mutants for *Pbx1* and *Pbx2* often show a smaller number of somites suggesting that these two genes could act redundantly in patterning the axial skeleton (*Capellini et al., 2008*). In *Pbx1−/−; Pbx2+/−* mutants, anterior shifts of *Hox* expression boundaries in the paraxial mesoderm have been reported (*Capellini et al., 2008*). Such shifts are consistent with a precocious ingression of cells normally fated to a more posterior identity.

Genetic studies on mouse T mutants have shown that graded T activity is required for body axis formation (*Stott et al., 1993*; *Wilson and Beddington, 1997*). Embryos with progressively lower quantities of T exhibit more severe axis truncations (*Stott et al., 1993*). Similar graded truncations are also observed for *Wnt3a* allelic series (*Galceran et al., 1999*), indicating that precise quantitative regulation of this pathway is required for completion of body axis elongation. Repression of the Wnt pathway and of T together with axis truncations was also observed in *Hox13* over-expressing transgenic mice (*Young et al., 2009*). Our data suggest that the gradient of T activity is established by the graded regulation of Wnt signaling by posterior *Hox* genes, (*Figure 11*) thus providing a possible explanation for these complex phenotypes. At the cellular level, it argues that the Hox-dependent regulation of T levels in the epiblast is critical to control the balance between cell ingression and maintenance of a self-renewing paraxial mesoderm progenitor pool in the epiblast/tail-bud. Cell ingression requires an EMT that involves destabilization of the basal microtubules of epiblast cells followed by basal membrane breakdown (*Nakaya et al., 2008*). Inhibiting Rhoa activity can rescue the ingression delay caused by *Hoxa13* overexpression, suggesting that posterior *Hox* genes can control cell flux to the PM by acting on basal microtubule stabilization in epiblast cells. As T is also able to rescue *Hoxa13* phenotype on elongation, it could act upstream of this process and the details of such a molecular pathway remain to be investigated.

Wnt signaling was proposed to promote the paraxial mesoderm fate at the expense of the neural fate in a population of bipotential neuro-mesodermal stem cells in the tail bud (*Martin and Kimelman, 2010*, *2012*; *Gouti et al., 2014*; *Tsakiridis et al., 2014*). Thus, the Wnt repression experienced by epiblast cells in response to posterior *Hox* genes overexpression could induce these cells toward a neural fate hence preventing them to ingress. However, *Hoxa13* overexpression does not lead to up-regulation of the neural marker *Sox2* in electroporated cells as detected by in situ hybridization and in our microarray analysis. Furthermore, electroporated cells are seen to enter the PM and do not enter the neural tube (*Figure 4* and supplementary videos). This therefore suggests that posterior *Hox* genes are unlikely to control cell ingression by promoting acquisition of a neural fate in epiblast cells. While we cannot completely rule out that a subpopulation of these cells remains in the tail-bud as *Sox2*-positive cells, it is unlikely that this contributes to the dramatic axis elongation slow down observed after posterior *Hox* genes overexpression.

Thus our data suggest that disruption of the balance between epiblast and ingressing cells can be achieved by interfering with T levels either directly or indirectly by altering Wnt levels or Hox expression. The graded repressive activity of posterior *Hox* genes on the Wnt/T pathway might provide an evolutionary constraint that led to the selection of collinearity of posterior *Hox* genes (*Duboule, 2007*).

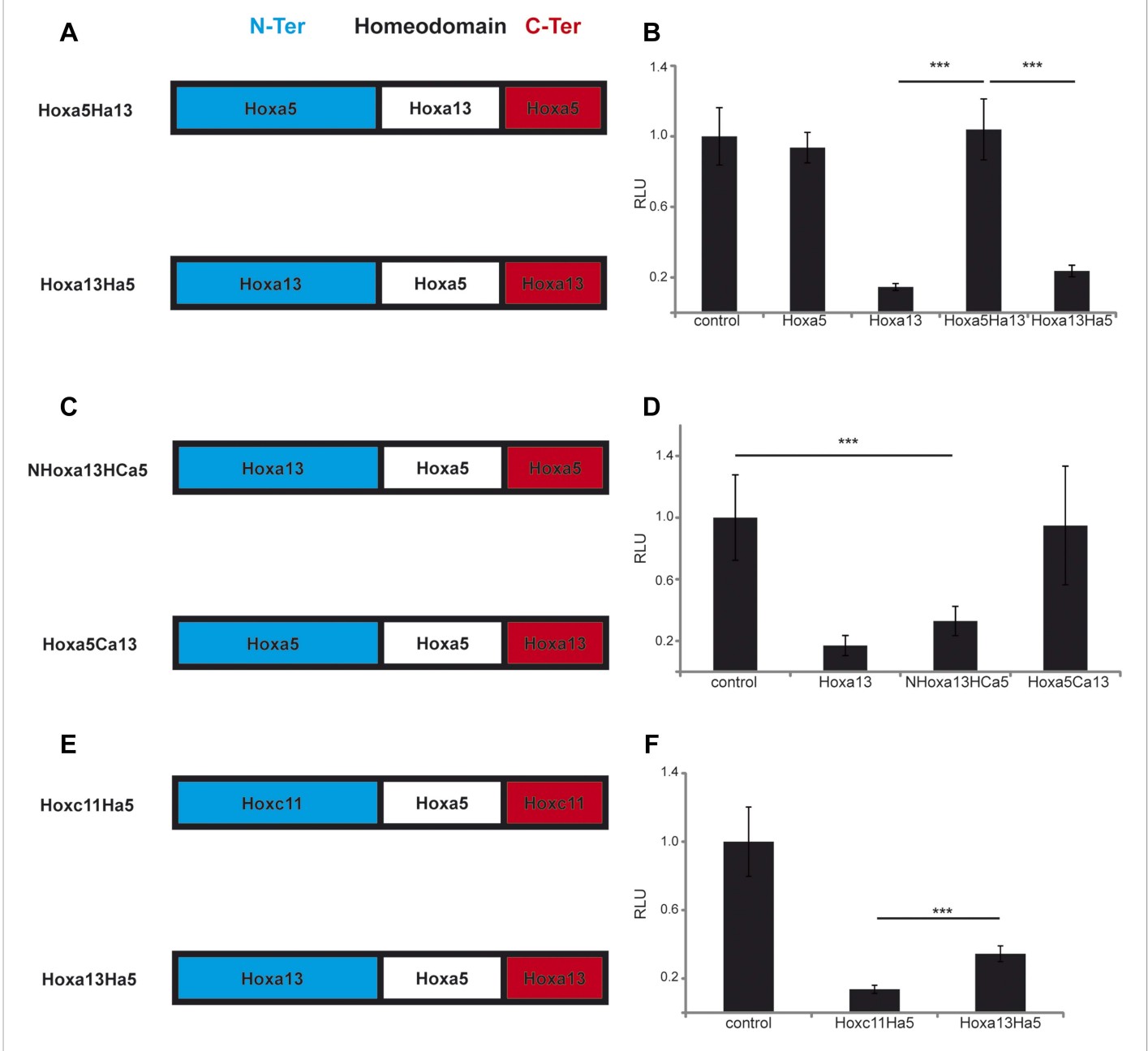

**Figure 10**. The N-terminal region of posterior Hox genes contains the repressive domain. (**A**, **C**, **E**) Design of the Hox chimeras. N-ter is in blue, the homeodomain in white, and the C-ter in red. (**B**, **D**, **F**) Luciferase assay measuring T/brachyury expression 20 hr after over-expression of cTprLuc and Renilla constructs together with (**B**) control, *Hoxa5*, *Hoxa13*, *Hoxa5Ha13*, or *Hoxa13Ha5*, (**D**) control, *Hoxa13*, *NHox13HCa5*, or *Hoxa5Ca13*, (**E**) control, *Hoxc11Ha5*, or *Hoxa13Ha5*. Stars represent the p-value of the two-tailed Student's t-test applied between the different conditions. ***p < 0.005. Error bars represent the standard error to the mean (SEM).

The following figure supplement is available for figure 10:

**Figure supplement 1**. The N-ter region of posterior Hox genes is poorly conserved at the amino-acid level.

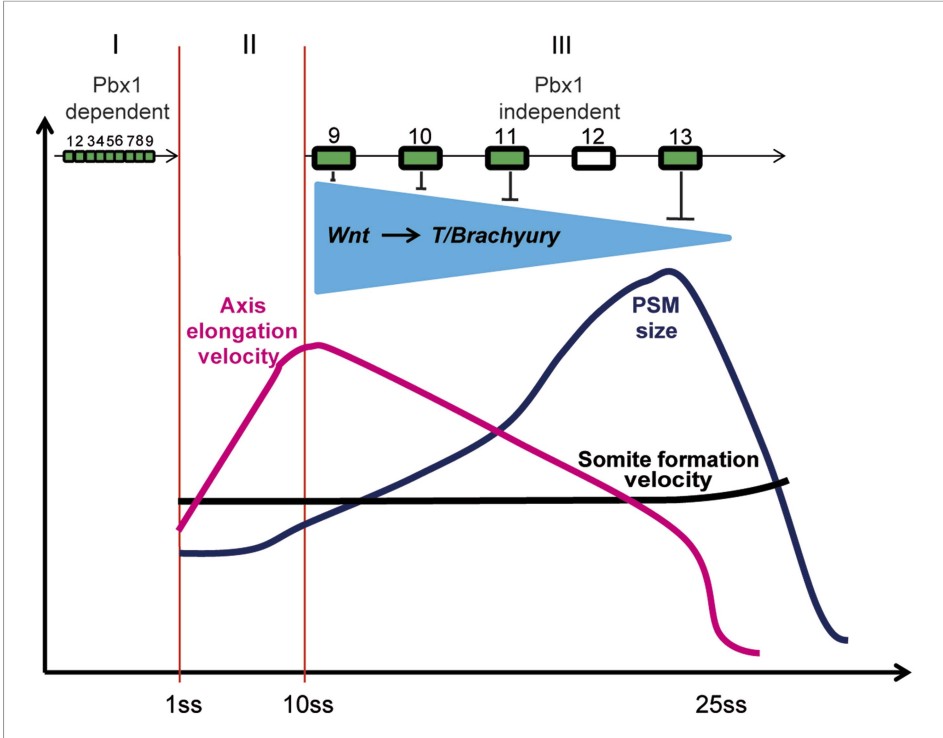

**Figure 11**. Model representing the 3 phases (I, II, and III) of *Hox* action in PM precursors in the epiblast/tail-bud during axis elongation. Model representing the 3 phases (I, II, and III) of *Hox* action in PSM precursors in the epiblast/tail-bud during body axis elongation. Anterior *Hox* genes (paralogs 1–9) are expressed during phase I. They control cell ingression in a *Pbx1*-dependent manner leading to the collinear positioning of *Hox* genes expression domains in the anterior region of the embryo. No *Hox* genes are activated during phase II, allowing fast elongation of the embryonic axis. During phase III, posterior *Hox* genes (paralogs 9–13) are collinearly activated in PSM precursors. Our data suggest that collinear activation of posterior *Hox* genes leads to repression of Wnt signaling and its target *T/Brachyury*, which progressively increases in strength. This results in a progressive arrest of cell ingression in the PSM, leading to a decrease in axis elongation rate. Since the velocity of somite formation is roughly constant, PSM size starts to decrease when elongation velocity becomes slower than that of somite formation. During this latter phase the control of cell ingression by posterior *Hox* genes appears to be independent of *Pbx1*.

## Materials and methods

### Chicken embryo culture ex ovo

Fertilized chicken eggs were obtained from commercial sources. Eggs were incubated at 38°C in a humidified incubator for approximately 24 hr. Embryos were prepared for Early Chick (EC) cultures (*Chapman et al., 2001*) and then electroporated. Embryos were staged following the Hamburger and Hamilton (HH) table (*Hamburger, 1992*) and by counting somites (somite stage: ss).

### Electroporation

Electroporation of the paraxial mesoderm (PM) precursors of the epiblast was carried out as described in *Bénazéraf et al. (2010)*. For the axis elongation assay, the electroporation success is first monitored 3 hr after by examining the embryos under a fluorescent stereomicroscope. Only embryos successfully electroporated in the PSM progenitors (90–100%) are processed for videos (see the axis elongation measurement section). For the luciferase assay embryos are electroporated in the PM progenitors and examined 20 hr after electroporation (see the Luciferase assay section). In both assays, we finally obtain 90–100% of the electroporated embryos showing reporter expression restricted to the paraxial mesoderm. In rare cases (less than 10%), we observed few cells in the lateral plate mesoderm. These embryos were discarded. To better illustrate the accuracy of the

electroporations performed, we now present a video showing the expression of a control construct electroporated in the anterior epiblast and showing the ingressing PSM cells (*Video 3*).

## Consecutive electroporation

To over-express two sets of constructs in different somitic precursors of the epiblast, two consecutive electroporations of the anterior primitive streak (PS) were carried out. Embryos were first prepared for EC culture at room temperature which pause their development. The first construct mixed with the control vector pCX-MyrCherry (gift from X Morin) was microinjected on one side of the PS groove and the first electroporation was carried out as described above right after injection. Only 10 s after the first electroporation, the second construct, containing the gene to over-express (*Hox*, *T*…) cloned in the pCAGGS-I2-MyrVenus, was then microinjected at the same level on the other side of the PS groove and immediately electroporated. This procedure targets the entire paraxial mesoderm territory of the epiblast of the anterior primitive streak on the electroporated side. Thus, in these experiments, we only track the timing of ingression of the most anterior epiblast cells which give rise to the most anteriorly localized progeny in the paraxial mesoderm. By biasing the position of the electrode on the right or on the left side of the primitive streak, we can ensure that the electroporation is biased on one side allowing easier observation of the anterior boundaries of the over-expressing cells. In most cases, however, some electroporated cells are seen on both sides despite this bias. The control Cherry vector is usually found to be more bilateral than the *Hox* expressing Venus-positive cells as can be seen from the pictures, which reflects an effect specific to *Hox* genes. Inverting the order of the electroporated plasmids in consecutive electroporation has no effect on the outcome of the experiment, and similar results are observed when the entire somitic territory of the epiblast of the anterior streak is electroporated. Following consecutive electroporation, embryos were cultured in a humidified incubator at 38°C to resume their development. 3 hr after electroporation, the embryos were screened based on fluorescence to ensure that both constructs were expressed and that the correct region was targeted for each construct. At this stage, up to 70% of the embryos are successfully electroporated. The embryos were then reincubated at 38°C until they reach the 15-somite stage (~20 hr). As can be seen on the embryos shown in *Figure 3*, *Figure 6*, and *Figure 9* or in the videos, virtually no cells are seen outside of the paraxial mesoderm meaning that the electroporation accurately targeted the anterior streak epiblast. Since expression of the constructs is driven by the ubiquitous CAGGS promoter, even a slight inaccuracy in positioning the electrode would result in expression in the neural tube or the lateral plate.

A fluorescent stereomicroscope (Leica M205 FA) equipped with a color camera was used to track anterior boundaries of both control Cherry-expressing cells and mutant Venus-expressing cells. At this stage, up to 60% of the electroporated embryos are exclusively electroporated in the PSM and retained for further analyses (~10% are discarded because of developmental defects due to the consecutive electroporation procedure). We used the 'Measure' plugin of ImageJ to measure the distance between the tail-bud and anterior boundary of both Venus and Cherry-expressing domains in the same embryos. We used these measures to calculate the ratio of Venus over Cherry domains. The Dot-Plot resulting from these ratios was generated using Graphpad 5 (Prism).

## Quantitative analysis of cell ingression

Paraxial Mesoderm (PM) progenitors in the anterior primitive streak were electroporated in Stage 5 HH embryos with either a control vector *pCAGGS-Venus* or *pCAGGS-Hoxa13-IRES2-Venus* and cultured in a humidified chamber at 38°C for 5 hr. Embryos were then fixed in 4% paraformaldehyde at room temperature for 40 min and immunolabeled for GFP, laminin, and DAPI as described below. Embryos were then mounted and imaged with a Zeiss 510 NLO equipped with a 20× NA 0.8 objective. 80 µm z-stack were acquired (one section every 0.42 µm), and cells were scored with respect to their position in the primitive streak or in the epiblast after 3D reconstruction and optical transverse sections using Imaris software. Epiblast cells were counted as 'non-ingressed' and primitive streak/mesodermal cells as 'ingressed'.

## RNA in situ hybridization and probes

Whole mount RNA in situ hybridizations were carried out as described (*Henrique et al., 1995*). Pictures of whole embryos were made using a macroscope (Z16APOA, Leica) with a 1× planapo objective (Leica) and a high resolution color camera (DFC 420C, Leica). Chicken *Pbx1* RNA probe is

described in *Coy and Borycki (2010)*. The *Fgf8* intronic probe is described in *Dubrulle and Pourquié, (2004)*. A 750-bp fragment of the coding sequence (from nucleotide 301 to 1061) of chicken *Fzd2*, the last 800 bp of the coding sequence of chicken *Dact2*, intron 6 of chicken *T*, a 948-bp fragment of *cSox2* coding sequence (gift from B Pain) were used as probes. Chicken *Hox* RNA probes were *Hoxa2* (*Prince and Lumsden, 1994*), *Hoxa3, b3, d4* (gift from R Krumlauf), *Hoxa10, a11, a13, c4, c5, c6, c8, c9, d8, d9, d10, d11, d12,* and *d13* (gift from C Tabin), *Hoxb1, b4, b7, b9* (described in *Iimura and Pourquié, 2006*), *Hoxb8* (gift from A Kuroiwa), *Hoxb5, c10, c11, c12, c13,* and *b13* (cloned by PCR using ENSEMBL sequence informations), *Hoxa4* (chEST 427p4, Geneservice), *Hoxa5* (chEST 382m24, Geneservice), *Hoxa6* (chEST 338h20, Geneservice), *Hoxa7* (chEST 259o11, Geneservice), *Hoxa9* (chEST 333f16, Geneservice), *Hoxb2* (chEST 194e4, Geneservice), *Hoxb6* (ChEST147L22, Geneservice), *Hoxd3* (chEST 195d1, Geneservice).

## Plasmid construction

Full-length coding sequences for chicken *Hox* genes, *Pbx1*, *T*, *Wnt3a*, *Wnt5a*, *Dact2*, and mouse *Fzd2* were PCR-amplified from chicken or mouse cDNAs using the proofreading *Accuprime pfx* DNA polymerase (Invitrogen, Grand Island, NY). PCR fragments were then cloned in either, Grand Island, NY P221 (Invitrogen) or pENTR-D/TOPO (Invitrogen) to generate gateway (Invitrogen) entry clones. The constitutively active version of *lef1* (βcatLEF) (gift from R Grosschedl) (*Galceran et al., 2001*), a dominant activated form of *Lrp6* (Lrp6ΔN) (gift from S Aaronson) (*Liu et al., 2003*), a stabilized form of *Ctbbn1* (dBC) (*Harada et al., 1999*), and a dominant negative form or *Rhoa* (DN-Rhoa) (Gift from P Kulesa) was PCR-amplified and sub-cloned in pENTR-D/TOPO (Invitrogen). *Hoxa13, Hoxc11,* and *Hoxd10* mutated versions unable to bind DNA (HoxmutH) were generated by mutating amino acids 50, 51, and 53 of the homeodomain to alanine (*Gehring et al., 1994*). When over-expressed in paraxial mesoderm precursors, these HoxmutH constructs show no effect on cell ingression, elongation velocity, and Wnt signaling (data not shown). The dominant-negative forms of *Hoxd10, c11,* and *a13* (respectively *Hoxd10dn, Hoxc11dn* and *Hoxa13dn*) were generated by inserting a stop codon instead of the amino acid 50 of the homeodomain. Chimeras for Hox genes were generated by fusion PCR. The homeodomain sequence of Hoxa13 was fused to the N-ter and C-ter of Hoxa5 to generate Hoxa5Ha13. The homeodomain of Hoxa5 was fused to the N-ter and C-ter of Hoxa13 to generate Hoxa13Ha5. The N-ter of Hoxa13 was fused to the homeodomain and C-ter of Hoxa5 to generate NHoxa13HCa5. The Cter of Hoxa13 was fused to the Nter and homeodomain of Hoxa5 to generate Hoxa5Ca13. The homeodomain of Hoxa5 was fused with either the Nter and Cter of Hoxc11 or the Nter and Cter of Hoxa13 to generate Hoxc11Ha5 and Hoxa13Ha5, respectively. The chimeras were cloned in pENTR-D/TOPO to generate entry clones. Entry clones were then cloned in destination vectors (depending on the experiments) using Gateway technology (Invitrogen).

For consecutive electroporations and luciferase assays, a pCAGGS-IRES2-Venus-RFA destination vector was generated as follows: a yellow fluorescent protein (YFP), Venus, with two sites of myristoylation that target the fluorescent protein to the membrane (Venus, gift from K Hadjantonakis) (*Rhee et al., 2006*), was fused to an Internal Ribosomal Entry Site (IRES2) (Clontech) by PCR. The primers used contained an EcoRI site in 5′ and a NotI site in 3′. The EcoRI/IRES2-Venus/ NotI fragment was then cloned into the EcoRI-NotI restriction sites of pCAGGS. A Gateway cassette (RFA, Invitrogen) was then inserted into the EcoRV site of the pCAGGS-I2-Venus, upstream of the IRES2.

For axis elongation measurements and cell tracking experiments, a pCI2HV-RFA destination vector was generated as follows: a YFP protein, Venus, was first fused to the full-length coding sequence of histone H2B to target the fluorescent protein to the nucleus (H2B-Venus). The H2B-Venus PCR fragment was then fused by PCR to an IRES2 (Clontech). The primers used contain an EcoRI site in 5′ and a NotI site in 3′. The EcoRI/IRES2-H2B-Venus/NotI fragment was then cloned into the EcoRI-NotI restriction sites of pCAGGS. A Gateway cassette (RFA, Invitrogen) was then cloned in the EcoRV site of the pCI2HV, upstream of the IRES2.

For luciferase assay experiments, the chicken *T* promoter (1 kb upstream of the ATG) was PCR-amplified and cloned upstream of the *firefly* luciferase in the pGL4.10 (luc2) vector (Promega) to generate the cTprLuc reporter. Expression driven by this promoter fragment in chicken embryo recapitulates the PM expression of *T* (not shown). The Wnt/βcatenin pathway activity reporter (seven TCF/LEF binding sites + *siamois* minimal promoter) was PCR amplified

from the BAT-GAL plasmid (Addgene plasmid 20889) (*Maretto et al., 2003*) and cloned upstream of a *firefly* luciferase in the pGL4.10(luc2) vector (Promega) to generate the BATLuc reporter.

## Pbx1 siRNA

RNA interference experiments were performed using 21-nucleotide dsRNAs (Dharmacon, Option A4). To identify electroporated cells, siRNAs (suspended in TE to a final concentration of 5 mg/ml) were mixed with a pCAGGS-Venus or Cherry expression plasmid (1.0 mg/ml). The target sequence against chick *Pbx1* was as follows: 5′- GTGTGAAATCAAAGAGAAA-3′. As a control siRNA, we used a siRNA targeting chick *Pbx1* containing two point mutations (underlined in the sequence): 5′-ACACAAAGCT-GAAGAAGTA-3′ that show no effect on *Pbx1* expression.

To monitor the *Pbx1* siRNA efficiency, the anterior primitive streak of stage 4 HH embryos was electroporated with either control siRNA or *Pbx1* siRNA mixed with a pCAGGS-Venus expression plasmid (1.0 mg/ml). Embryos were reincubated at 38°C until they reach stage 7 HH when they were harvested and processed for ISH for *Pbx1* and immunofluorescence against GFP.

## *Pbx1* over-expression

A pBIC control vector (derived from the pBI-tet [clontech] in which Cherry has been cloned) (gift from J Chal) that allows simultaneous expression of two proteins at the same level once activated by doxycycline (Tet-on, Clontech) or the pBIC vector containing the full length *Pbx1* along with a vector expressing the rtTA (Clontech) were electroporated in PSM progenitors at Stage 5 HH. Embryos were reincubated until they reach the 3-somite stage. They were then placed on imaging plates containing 0.5 µg/ml doxycycline for 1 hr at 38°C before starting acquisition. Axis elongation measurements were performed as described below between 5 and 9-somite stages.

## Time-lapse microscopy

Electroporated embryos were cultured ventral side up on a microscope stage. We used a computer controlled, wide-field (10× objective) epifluorescent microscope (Leica DMR) workstation, equipped with a motorized stage and cooled digital camera (QImaging Retiga 1300i), to acquire 12-bit grayscale intensity images (492 × 652 pixels). For one embryo, several images at different focal planes and different fields were captured at a single time-point (frame). The acquisition rate used was 10 frames per hour (6 min between frames). Image processing, including focal plane 'collapsing' field merging and registering, was performed to create high-resolution, 2D time-lapse sequences for cell tracking and axis elongation measures (see *Czirók et al., 2002*, for details). To correct for the gradual drift of the embryo position or sudden changes due to repositioning of the microscope stage, images were registered to the embryo center.

## Axis elongation measurements

Variation of the distance between a formed somite and the node was used to determine the velocity of body axis elongation. The coordinates of the different points were determined on bright-field images of the time-lapse experiments using the cellular tracking option of ImageJ. ImageJ is a public domain, Java-based image processing program developed at the National Institutes of Health.

For wild-type embryo measurements, axis elongation velocity was measured between 1 and 3 somites (n = 8), between 5 and 7 somites (n = 8), between 9 and 11 somites (n = 6), between 15 and 17 somites (n = 5), between 20 and 22 somites (n = 6), and between 25 and 27 somites (n = 8). For 15–17 somites measurements, embryos were cultured starting at 13 somites and imaged until 18 somites. For 20–22 somite measurements, embryos were cultured starting at 18 somites and imaged until 23 somites. For 25–27 somite measurements, embryos were cultured starting at 23 somites and imaged until 28 somites. For measurements of axis elongation velocity after *Hox* or *T* over-expression, electroporated embryos at stage 5 HH were cultured in a humidified incubator at 38°C for 3 hr and then placed on the microscope stage, as described above, for 18 hr. Axis elongation velocity was measured for 10 hr, starting from the 5-somite stage. Student's t-tests were applied to evaluate the differences between conditions.

## Cell tracking

Cells electroporated with either a control or a *Hox* gene and a nuclear fluorescent protein (H2B-Venus or H2B-GFP) were automatically tracked using the Imaris software's cell tracking module (version 7.3.1). Cells were segmented based on nucleus size (set at 5 µm) and fluorescence intensity. The tracking algorithm was based on Brownian motion. Only cells in the posterior PSM were tracked for 10 hr. To substract the tissue motion to the single cell motion, the average speed of all tracked cells, that represent the tissue motion, has been substracted from the average speed of each individual cell (as described in *Bénazéraf et al., 2010*). Student t-tests were applied to evaluate the differences recorded between the different conditions.

## Luciferase assay

Embryos were harvested at stage 5 HH and electroporated with a DNA mix containing either cTprLuc or BATLuc (1 µg/µl final), CMV-Renilla (Promega, Madison, WI) (used as a control to normalize the differences of electroporation intensity between embryos [0.2 µg/µl final]), a control pCAGGS-Venus vector (gift from K Hadjantonakis) or a gene of interest cloned in pCAGGS-IRES2-Venus (5 µg/µ; final). Electroporated embryos were cultured in a humidified incubator at 38°C for 20 hr. Embryos were analyzed using a fluorescent microscope and only embryos showing restricted expression of Venus in the paraxial mesoderm were selected (90–100% of the electroporated embryos) for luciferase assay (between 3 and 5 embryos for each condition). The posterior region (from somite 1 to tail-bud) of the selected embryos was dissected and lysed in passive lysis buffer (Promega) for 15 min at room temperature. Lysates were then distributed in a 96-well plate and luciferase assays were performed using a Centro LB 960 luminometer (Berthold Technology, France) and the dual luciferase kit (Promega) following manufacturer's instructions. Raw intensity values for Firefly luciferase signal were normalized with corresponding Renilla luciferase values (RLU) and the control experiment was set to 1. Student t-tests were applied to evaluate the differences between conditions.

For *Hox* dominant-negative experiment, embryos were electroporated at st8 HH with a mix containing BATLuc, CMV-Renilla and either a *Hoxa13mutH* or a mix of *Hoxc11mutH* and *Hoxa13mutH* or a mix *Hoxd10mutH*, *Hoxc11mutH* and *Hoxa13mutH* (in pCAGGS-I2-Venus [control condition]) or *Hoxa13dn*, or a mix of *Hoxc11dn* and *Hoxa13dn* or a mix of *Hoxd10dn, Hoxc11dn* and *Hoxa13dn* (in pCAGGS-I2-Venus [mutant condition]). Embryos were reincubated until they reach the 28-somite stage. The tail-bud of each embryo was dissected and used for the luciferase assay as described above.

## Histology, immunohistochemistry, and imaging

Stage 5 HH embryos electroporated with either a control pCI2HV or a pCI2HV*Hoxa13* vector were cultured in a humidified incubator at 38°C for 6 hr. Embryos were then selected using a fluorescence stereomicroscope based on electroporation efficiency. Selected embryos were fixed for 30 min at room temperature and then cryo-preserved in 30% sucrose in PBS at 4°C. Embryos were then transferred in a solution containing 7.5% gelatin and 15% sucrose in PBS and placed at 42°C. Embryos were then included in a cryosection mold and flash frozen in a dry ice-ethanol bath. 12-µm transverse cryosections of the electroporated region were prepared using a Leica CM3050 S cryostat. Sections were collected on superfrost slides and stored at −20°C. For immunocytochemistry, sections were placed in warm PBS (42°C) for 5 min to remove gelatin. Sections were incubated with the primary antibody in PBS/BSA (2%)/Triton (0.1%) for 2 hr in a humidified chamber at room temperature. Slides were then washed four times for 15 min in PBS and incubated with the secondary antibody in PBS/BSA (2%)/Triton (0.1%) for 45 min in a humidified chamber.

For the cell ingression assay and the labeling of the extracellular matrix (ECM), we used, respectively, a rabbit anti-GFP (abcam, #ab290, UK) at 1/2000 and the mouse anti-laminin (DSHB, #3H11, Iowa City, IA) at 1/200. The secondary antibodies were anti-rabbit Alexafluor488 (Invitrogen) and anti-mouse IgG1Alexafluor555 (Invitrogen), respectively, used at 1/1000. DAPI (Invitrogen, 1/1000 dilution) and an Alexafluor 633 phalloidin (Invitrogen) were applied at the same time as the secondary antibodies to label the nuclei and the F-actin, respectively. For tubulin labeling, we used the mouse anti-acetylated alpha-tubulin (sigma T6793) at 1/1000. The secondary antibody was an anti-mouse IgG2b Alexafluor546 (Invitrogen), used at 1/1000. Slides were mounted in Fluoromount-G

(SouthernBiotech) and analyzed with a LSM 510 NLO inverted confocal microscope (Carl ZEISS, Germany) using a plan apochromat 63× (NA 1.4) immersion (oil) objective (Carl Zeiss).

## Hoxa13 protein quantification

A pBIC control vector (described above) or the pBIC vector containing the full-length *Hoxa13* with a C-terminal HA tag along with a vector containing EGFP under the CAGGS promoter and a vector expressing the rtTA (Clontech, France) were electroporated in the PM progenitors at stage 5 HH. A drop of 50 µl of different doses of doxycyclin (from 50 µg/ml to 0.5 µg/ml) was applied on top of the embryos immediately after electroporation and the embryos were reincubated for 20 hr. Three embryos for each condition were individually lysed following standard procedure and each lysate was loaded on a different well of an SDS-page gel. Western blot analysis was done following standard procedure. An anti-HA-HRP antibody was used to detect Hoxa13 (Roche #12013819001, dilution 1/1000, Germany). An anti-GFP antibody (abcam ab6556, dilution 1/2000) was used to detect GFP from the pCAGGS-EGFP used as an electroporation control. An anti-β actin antibody (Sigma A5441, dilution 1/5000, Germany) was used to verify that the same amount of tissue was loaded in each well. This experiment has been repeated twice independently.

## Cell proliferation analysis

A 20 µl drop of 100 µM EdU (Click-iT EdU kit, Cat. #C10083 Invitrogen) was applied on the posterior region of 20–22- and 25–27-somite stage embryos cultured in vitro for 45 min. Embryos were then immediately fixed in 4% paraformaldehyde (PFA) for 45 min at room temperature (RT) and were then processed as described in *Warren et al. (2009)*. Phospho-histone H3 (pH3) (Millipore, #06-570, 1/1000 dilution, France) immunolabelling was performed after the EdU reaction. Single plane sections were generated, and the PSM region was manually segmented. For the tail-bud proliferation assay, parasagittal cryosections (20 µm) were made. Nuclei labeled with DAPI and EdU and/or pH3 were manually counted. Sections were imaged using a Zeiss 510 NLO and a 20× dry NA0.8 objective.

## Apoptosis quantification

Embryos were harvested at 20–22- and 25–27-somite stage and processed as described (*Smith and Cartwright, 1997*) using the ApopTag Red In Situ kit (#S7165; Millipore). Single plane sections were generated and the PSM region was manually segmented. For tail-bud apoptosis assay, parasagittal cryosections (20 µm) were made. Nuclei labeled with DAPI and/or apoptotic labeling were manually counted. Labeled embryos were imaged using a Zeiss 510 NLO and a 20× dry NA0.8 objective.

## Microarray analysis

PM precursors of the anterior primitive streak of Stage 5 HH embryos were electroporated as previously described either with a control vector coding for a *H2B-venus* fusion (pCI2HV) or a vector coding for *Hoxa13* and a *H2B-venus* fusion (*Hoxa13*pCI2HV). Embryos were reincubated for 14 hr in a humidified incubator at 38°C until they reach the 9-somite stage. The region containing the PM progenitors was dissected from several embryos and pooled in a drop of PBS/FCS1% (seven embryos per condition) on ice. Dissected tissues were then transferred in a drop of diluted trypsin and incubated at 38°C for 10 min to allow efficient enzymatic dissociation of cells. Cell dissociation was completed mechanically by pipetting up and down. Cells were then transferred into 500 µl of PBS/FCS 1% on ice and sorted based on YFP fluorescence using a FACS DIVA (BD technologies, France). For each condition, one thousand YFP+ cells were collected directly in Trizol (Invitrogen) and immediately frozen at −80°C. This experiment was repeated twice independently.

Extraction of total RNA was performed according to manufacturer's instructions (Trizol, Invitrogen). Biotinylated cRNA targets were prepared from total RNA using a double amplification protocol according to the GeneChip Expression Analysis Technical Manual: two-Cycle Target Labeling Assay (P/N 701021 Rev.5, Affymetrix, Santa Clara, USA). Following fragmentation, cRNAs were hybridized for 16 hr at 45°C on GeneChip Chicken Genome arrays. Each microarray (one microarray per condition = two control microarrays and 2 *Hoxa13* microarrays) was then washed and stained on a GeneChip fluidics station 450 and scanned with a GeneChip Scanner 3000 7G. Finally, raw data (.CEL Intensity files) were extracted from the scanned images using the Affymetrix GeneChip Command Console (AGCC) version 3.1. CEL files were further processed with MAS5 and RMA algorithms using the Bioconductor package (version 2.8) available through R

(version 2.12.1). Probe sets were filtered based on their expression intensity value (MAS5 value). Probe sets with an intensity value under 100 were discarded. Probe sets were ranked based on fold change between the intensity value of the control condition and the *Hoxa13* over-expression condition. The microarrays raw data are available on the GEO website (http://www.ncbi.nlm.nih.gov/geo/query/acc.cgi?acc=GSE38107).

## Q-RT PCR analysis of FACS-sorted cells over-expressing *Hoxa13*

RNAs from the microarray experiments were used as templates for cDNA synthesis using the Qantitect kit (Qiagen). 3 µl of cDNA was mixed with 6 µl of 2× Lightcycler 480 SYBR green I master (Roche) and 1 µM of primers (listed in *Table 2*) in a total volume of 12 µl. The Q-PCR reactions were run on a Lightcycler 480 (Roche) with the Lightcycler 480. Each sample was run in duplicate and *gapdh* was used as a control gene. The CT values obtained for each gene were normalized against the CT value obtained for *gapdh*.

## Q-RT PCR analysis of FACS-sorted cells over-expressing the dominant-negative constructs

PM precursors of the anterior primitive streak of Stage 8 HH embryos were electroporated as previously described either with a mix of control vectors coding for *Hoxd10mutH*, *Hoxc11mutH* and *Hoxa13mutH* in pCAGGS-IRES2-Venus or a mix of vector coding for *Hoxd10dn*, *Hoxc11dn* and *Hoxa13dn* in pCAGGS-IRES2-Venus. Embryos were reincubated in a humidified incubator at 38°C until they reached the 28-somite stage. Tail-bud regions containing the PM progenitors were then dissected and pooled in a drop of PBS/FCS1% (three embryos per condition) on ice. Dissected tissues were then transferred into a drop of diluted trypsin and incubated at 38°C for 10 min to allow efficient dissociation of the cells. The dissociation of cells was completed mechanically using a glass micropipette by pipetting up and down. Cells were then transferred into 500 µl of PBS/FCS1% on ice

**Table 2**. List of primers used for Q-RT PCR

| Gene name | Gene reference | Primers sequence 5′→3′ | Size of the amplicon |
|---|---|---|---|
| *Gapdh* | NM_204305.1 | F: GCTGAGAACGGGAAACTTGTG | 62 bp |
| | | R: GGGTCACGCTCCTGGAAGA | |
| *T* | NM_204940.1 | F: CGAGGAGATCACAGCTTTAAAAATT | 75 bp |
| | | R: TCATTTCTTTCCTTTGCGTCAA | |
| *Axin2* | NM_204491.1 | F: GCGCAAACGATAGTGAGATATCC | 76 bp |
| | | R: CCATCTACACTGCTGTCTGTCATTG | |
| *Sp8* | NM_001198666.1 | F: CATGGCGCACCCCTACGAGTC | 131 bp |
| | | R: CGTTGGGGGCACGTCGATCCA | |
| *Fzd2* | NM_204222.1 | F: CCCTGCCCGCTGCACTTCAC | 190 bp |
| | | R: CCGCTCACACCGTGGTCTCG | |
| *Cyp26a1* | NM_001001129.1ik | F: AGGAGCCCGAGGGTGGCTACA | 138 bp |
| | | R: TGGCAGTGGTTTCATGACCTCCAA | |
| *Fgf8* | NM_001012767.1 | F: CGCTCTTCAGCTACGTGTTCATGC | 108 bp |
| | | R: TGGTAGGTGCGCACGAGCC | |
| *Etv1* | NM_204917.1 | F: ATGGACCACAGATTTCGCCGCC | 145 bp |
| | | R: TTGGACGTCCTTCCCTCGGCA | |
| *Fgfr1* | NM_205510.1 | F: CACGCTGCCCGACCAAGCTC | 168 bp |
| | | R: GTGATGCGCGTGCGGTTGTT | |
| *Rasgrp3* | NM_001006401.1 | F: AACGGCATCTCCAAGTGGGTCCA | 111 bp |
| | | R: GAGATGAAGGAGCTTCTGTGCAACA | |

and sorted based on YFP fluorescence using a FACS DIVA (BD technologies). For each condition, one thousand YFP+ cells were collected directly in Trizol (Invitrogen) and immediately frozen at −80°C. This experiment was repeated four times independently.

Extraction of total RNA was performed according to manufacturer's instructions (Trizol, Invitrogen). RNAs were used as templates for cDNA synthesis using the QuantiTect kit (Qiagen). 3 μl of cDNA was mixed with 6 μl of 2× Lightcycler 480 SYBR green I master (Roche) and 1 μM of primers (listed in *Table 2*) in a final volume of 12 μl. The Q-RT PCR reactions were run on a Lightcycler 480 (Roche). Each sample was run in duplicates, and *gapdh* was used as a control. The CT values obtained for each gene were normalized against the CT value obtained for *gapdh*.

## Q-RT PCR analysis of microdissected tailbuds

Embryos were harvested in PBS at different stages (10, 15, 20, or 25-somite stages) and pined using 0.10 mm minutiens on a silicon-coated petri dish. The tailbud region was then microdissected using a sharpened tungsten needle and care was taken to remove the endoderm and the ectoderm. Each individual tailbud was immediately transferred in 500 μl of Trizol (Invitrogen) in a 1.5 ml RNAse free tube (Ambion) on ice until five individual tailbuds were collected per stage (resulting in five tubes per stage). Then the tubes were immediately frozen at −80°C. Extraction of total RNA was performed according to manufacturer's instructions (Trizol, Invitrogen). RNAs were used as templates for cDNA synthesis using the iScript reverse transcriptase Supermix (Biorad). 3 μl of cDNA was mixed with 5 μl of 2× SSoAdvanced universal SYBR green supermix (Biorad) and 1 μM of primers (listed in *Table 2*) in a final volume of 10 μl. The Q-RT PCR reactions were run on a CFX384 (Biorad). Each sample was run in triplicates, and *gapdh* was used as a control. The CT values obtained for each gene were normalized against the CT value obtained for *gapdh*.

## Acknowledgements

The authors thank V Wilson, D Wellik, L Selleri, and members of the Pourquié laboratory for comments. The authors acknowledge P François, P Moncuquet, C Ebel, and O Tassy for help. This research was supported by the Howard Hughes Medical Institute, the Stowers Institute for Medical Research, a NIH grant R02 HD043158 and a Chaire d'excellence of the Agence Nationale pour la Recherche (ANR) as well as the European Research Council.

## Additional information

### Funding

| Funder | Grant reference number | Author |
|---|---|---|
| Howard Hughes Medical Institute (HHMI) | | Olivier Pourquié |
| Stowers Institute for Medical Research | | Nicolas Denans, Tadahiro Iimura, Olivier Pourquié |
| National Institutes of Health (NIH) | R02 HD043158 | Olivier Pourquié |
| Agence Nationale de la Recherche | | Olivier Pourquié |

The funders had no role in study design, data collection and interpretation, or the decision to submit the work for publication.

## Additional files

### Supplementary file

• Supplementary file 1. List of genes regulated by *Hoxa13* in our microarray screen. (**A**) List of genes upregulated in the PSM precursors after Hoxa13 overexpression. (**B**) List of genes downregulated in the PSM precursors after Hoxa13 overexpression.

## Major dataset

The following previously published dataset was used:

| Author(s) | Year | Dataset title | Dataset ID and/or URL | Database, license, and accessibility information |
|---|---|---|---|---|
| Denans N, Moncuquet P | 2012 | HoxA13 gain-of-function in chicken embryo PSM progenitors | GSE38107 | Publicly available at NCBI Gene Expression Omnibus (http://www.ncbi.nlm.nih.gov/geo/). |

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
