## [Decision Letter]

Thank you for sending your work entitled “*Hox* genes control
vertebrate body elongation by collinear Wnt repression” for consideration at
*eLife*. Your article has been favorably evaluated by Janet Rossant
(Senior editor), Marianne Bronner (Reviewing editor), and three reviewers.

The Reviewing editor and the three reviewers discussed the comments before we reached
this decision, and the Reviewing editor has assembled the following comments. It is the
policy of *eLife* to only ask for revisions that can be accomplished in a
reasonable time. Given the extensive revisions that would be required to satisfy the
reviewers' concerns, we cannot accept your paper for publication at this time.
However, we would be open to considering a new submission at a later time that addresses
the reviewers’ comments after completion of further experiments and addition of
extensive revisions to the text.

Summary:

The manuscript by Denans, Iimura and Pourquie presents evidence for a role of
*Hox* genes in mesoderm ingression and body elongation in vertebrates.
They test the interesting hypothesis that some posterior *Hox* genes
repress Wnt and FGF signaling and so decrease expression of mesodermal gene T, which
then leads to reduced ingression of cells through the primitive streak. This eventually
leads to termination of the body elongation process by lack of available mesoderm. In
this view, 'posterior *Hox* genes' regulate the extension
of the body plan by indirectly controlling the amount of available mesoderm cells in the
tail end of the embryo (via the regulation of cell ingression from the epiblast).

This is an interesting piece of work, which adds important arguments supporting the
function of *Hox* genes during the elongation of the vertebrate body
axis. However, the reviewers raised major concerns regarding technical approaches, the
lack molecular explanation for the described phenomena, and appropriate reconciliation
of the authors’ conclusions with previously published work. Additional control
experiments and further discussion of the literature as suggested below are required to
substantiate if the proposed mechanism indeed holds true.

1) One concern is the quantification of all the various phenomenon described in this
work. There are several parameters, which can easily escape control in this context
(mRNA stability, protein presence, electroporation success, copy number, saturation
etc.). This criticism is not particular to this work and a similar question could be
asked regarding the results by others in mice in mice, where the strong effect was not
explicitly related to a 'physiological' gain of function (even though it
may indeed be the case). Vertebral formula is very stable within the same species and
thus must rely upon a robust mechanism. How the robustness of this process relates to
the quantities of factors is a key issue (are these factors saturating?) At the time of
the start of decreasing Wnt signaling due to increasing Hox amounts, a rather smooth
gradient of T must be established. At these time points, genetic conditions in mice
would be expected to have an effect perhaps stronger than what was reported by Capecchi
for the *Hoxb13* mutants? And if several *Hox* genes
contribute to this phenomenon, then how to prevent intra-species variations, in
particular at the time-points when the system would no longer be saturated?

2) From this model, one can make rather strong predictions, such as e.g. that the
addition of yet another Hox cluster with posterior genes should make the body shorter?
Is this the case in the literature?

3) There are important differences in effects between the *Hoxa13* and
*Hoxd13* genes. In many developmental contexts, these two genes have
very redundant functions and therefore, the difference reported here would suggest that
Hoxa13 achieves its function in the tailbud by affecting a pathway or by using a peptide
sequence that is not associated or existing in *Hoxd13*? This is
particularly unexpected, as *Hoxd13* (at least in
mammals) seems to be expressed in the late tail bud more strongly than
*Hoxa13*. This should be discussed.

4) A major claim of the authors is that there is delayed ingression, but this is not
convincing. They report delayed expression of the Hox-Ires-Venus cassettes following
electroporation into the chick epiblast as measured by Venus protein fluorescence. The
authors interpret this effect such that *Hox* gene over-expression can
delay ingression of the cells receiving the construct. However, the Hox constructs are
driven by the ubiquitous Caggs promoter. Why is expression of the Hox constructs not
observed at the same time as that of the control construct? A simple explanation might
be that the RNA derived from these constructs is post-transcriptionally controlled,
either by immediate degradation or by translational repression. This might assure
position-adequate activity of the *Hox* genes. RNA in situ hybridization
experiments testing for the time and position of transcription of the constructs are
missing. However, they are required at least for some of the constructs. In addition,
co-transfection of the control reporter construct with the Hox expression constructs
should reveal if indeed ingression is delayed by Hox over-expression. Furthermore, RNAi
of critical *Hox* genes (e.g. *Hoxa13*) would be a means
of testing if the proposed effect of *Hox* genes can be opposed.

5) Critical missing data are the level of Brachyury expression along the body axis. If
the authors are right that increasing *Hox* gene expression has a
repressive effect on Brachyury expression resulting in decreasing mesoderm formation and
motility along the body axis, this should be reflected by decreasing Brachyury
transcription in the wild-type embryo as it elongates. Is this the case? Of course this
would mean measuring expression levels per mesoderm cell (e.g. FPKM or qPCR values
determined on FAC sorted cells) at different embryonic stages but at the same axial
position (e.g. close to the node), not just overall expression in the embryo. The
authors have all the tools at hand for doing the test.

6) It is not clear if the data from chick can be extrapolated to vertebrates in general.
If the proposed mechanism were generally true, vertebrates with tails should not exist.
All *Hox* genes are already activated during trunk development and the
tail does not gain extra Hox activity grinding Brachyury expression eventually down to a
standstill. Thus in tailed animals a different mechanism is more likely. The authors
therefore should present their conclusions with respect to the chick and possibly other
vertebrates with tiny tails (humans, some monkeys?).

7) The accuracy of the experimental approaches is difficult to assess, in particular,
the accurate and reproducible targeting of the epiblast layer of the primitive streak by
electroporation is not documented. It seems likely that there is some variability
between embryos and that in some cases cells that are already ingressing are targeted.
Furthermore, it seems possible that control cells are sometimes affected by Hox
mis-expressing cells in the primitive streak, making this assay difficult to interpret
with confidence, particularly when only a handful of embryos are analyzed for each gene.
A data set demonstrating localized epiblast targeting reproducibility would strengthen
the claims of the authors that they are assessing regulation of the ability of
mis-expressing cells to ingress. It is also concerning that the arrow indicating the
most rostral red fluorescent label in Figure 3
is mis-placed caudally.

8) The time difference between “successive” electroporations should be
clarified; if the first construct labels cells earlier then these will progress to a
more rostral position than cells electroporated slightly later, which could result in a
misleading conclusion that the second construct used delays ingression. It seems the
control construct is always electroporated first; can the authors invert the order in
which the constructs that induce a delay are electroporated and obtain the same
result?

9) There is no explanation at a molecular level for differences in the ability of
posterior *Hox* genes to delay ingression nor for the effect of posterior
dominance within the subset of posterior *Hox* genes that appear to
affect ingression. How does collinear onset of posterior *Hox* genes lead
to progressive loss of Wnt/FGF and T if this does not involve protein accumulation?
Identifying these molecular mechanisms is needed to take the paper beyond the discovery
of an interesting phenomenon.

10) We are left at the end of the paper without a molecular explanation for how
*Hox* genes influence the ingression of cells. It is surprising that a
gene that leads to reduced Wnt and FGF signaling in primitive streak epiblast does not
affect cell adhesion and E-cadherin expression (data for which is not shown) nor the
balance of *Sox2/T* expression in this cell population which would lead
to cells adopting a neural over a mesodermal fate. Previous work has shown that FGF
signaling is upstream of E-cadherin downregulation, prompts EMT and formation of T
expressing mesoderm at the mouse primitive streak (Ciruna and Rossant, 2001, Dev. Cell),
while more recent work has demonstrated that high Wnt signaling promotes mesoderm
differentiation from neuro-mesodermal (*Sox2/T* co-expressing)
progenitors and that reduced Wnt signaling in this context leads to neural
differentiation ([67], Dev;
[27], PLOS Biology). Other
work, in chick, analyzing the expression of *Sox2* and T transcription
throughout body axis elongation shows that strong T persists in the tailbud long after
the 25 somite stage, declining only just prior to cessation of elongation at ∼ 44
somite after which T begins to decline and *Sox2* is up regulated in
tailbud cells, in a step that involves decreased FGF signaling ([52], PLOS Biology).

Indeed, [66], BMC Dev. Biol. show
that elongation cessation takes place when the presomitic mesoderm is still present at
HH24 (rather than having been used up by somites forming right to the tail tip),
suggesting that the body axis does not end due to a lack of ingressed cells. This
observation contradicts the main conclusion of this current paper that it is a lack of
ingressed presomitic cells that underlies elongation arrest.

11) The authors present only *Sox2* in situ data (Figure 2–figure
supplement 2) in whole mount embryos to support their claim that there is not a cell
fate change when *Hoxa13* is mis-expressed. Sections of such embryos in
which the *Hoxa13* GFP construct is visible with the
*Sox2* in situ signal on a cell by cell basis is required for
convincing evidence on this point. A further possibility is that *Hoxa13*
expressing cells simply down regulate T but remain *Sox2* positive and so
adopt a neural fate and therefore do not ingress. Again comparing control and
*Hoxa13* GFP positive cells for levels of T and *Sox2*
on a cell-by-cell basis may provide evidence for the decline in T and maintenance
*Sox2*.

12) It is not clear why the authors provide in situ images for *T* and
*Fgf8* at HH14 (Figure 5) using
an intronic probe; other genes are not assessed in this way and no comment or
explanation is provided in the text or figure legend. It is not clear how this data
compares with published work showing strong and extensive T transcripts until the end of
axis elongation. The authors need to address this difference with previous work, explain
the significance of loss of active T transcription and consider how T transcripts
continue to be regulated by FGF in the tailbud at these late stages (52) if the gene is not
actively transcribed. One logic may be that the effects of endogenous posterior
*Hox* genes are very small and slow acting, however, the authors
appear to argue that there is a step change as the tailbud forms and the image presented
in Figure 5 implies that T transcripts are lost
in the tailbud at this time.

[Editors' note: further revisions were requested prior to acceptance, as
described below.]

Thank you for resubmitting your work entitled “*Hox* genes control
vertebrate body elongation by collinear Wnt repression” for further consideration
at *eLife*. Your revised article has been favorably evaluated by Janet
Rossant (Senior editor), Marianne Bronner (Reviewing editor), and the original three
reviewers. The manuscript has been greatly improved and is acceptable pending changes
that can easily be accomplished by revisions in writing of the manuscript without
requiring further experiments. However, the reviewers feel that being more cautious in
interpretation is important so we ask you to carefully attend to these remaining issues
that need to be addressed before acceptance, as outlined below:

Overall, this paper presents a striking phenomenon in the regulation of cell ingression
through the primitive streak by posterior *Hox* genes and some very
interesting data that explain to some extent the molecular mechanisms at play here. The
authors still need to place the work in an appropriate context both in terms of the
literature and the significance of this mechanism during body axis elongation; it may be
that collinear activation of posterior *Hox* genes regulates tailbud
morphogenesis half way through axis elongation, but does not directly regulate PSM
length.

1) The role of RhoA is not sufficiently well investigated and documented for giving it a
prominent role in the Abstract. If the authors find this point so important they should
provide some more experimental support. Alternatively they may want to rewrite the
Abstract.

2) Last line of the Abstract (end of the Introduction and elsewhere,): “this
mechanism leads to progressive reduction of PSM size associated to the arrest of
somitogenesis” does not follow logically. What does this mean if it is clear
that, at least in the chick, body axis elongation ceases when the PSM is still present
and there is room for more somites to be generated? How is ingression then associated
with arrest of somitogenesis? Somitogenesis arrest is more likely linked to loss of
Notch pathway oscillations in the remaining PSM (66).

3) In the Introduction the authors should mention the evidence that explanted chick
tailbuds (i.e. without somites) are a demonstrated source of RA (66). This is an important finding, which bears on
their hypothesis that shortening of the PSM is a major step in that brings retinoic acid
to tailbud and so curtails body axis elongation. It may be that both mechanisms operate
and I think this should be made clear upfront, rather than appearing later in the
Discussion.

4) In the Introduction and elsewhere it is confusing to state that
“*Hoxb* genes are only expressed in anterior regions of the
embryo precluding their playing a role in the control of axis extension”, when
*Hoxb13* is known to regulate axis elongation. They need to be more
precise, *Hoxb1-9*.

5) It would be informative to explain in the paper that the 25 somite stage is
∼when the tailbud forms and that this is when when cell ingression through the
primitive streak ceases—refer to work from Susan Mackem's lab in chick
showing cessation at HH stage16 (∼ 24-28 somites, [37] and see [9], and also [73], in the mouse). It seems likely that the mechanism identified
by the authors is most functionally significant at this mid-way stage; from then on the
PSM is added to from an “internalized” set of precursor cells. McGrew et
al, 2009, and [52]
identify these cell populations by lineage analysis in the chick tail bud as the NMps
(Sox/bra expressing cells) and also a transit amplifying cell population of mesodermal
progenitors. I think the authors should explain this in the Discussion (and in the
Introduction). As the paper currently reads it might be thought that ingression through
the primitive streak continues until the end of axis elongation, when in fact in both
chick and mouse embryos this ceases at somite 30 of ∼50-53 (chick, HH16) and 33
of 65 (mouse, E10.5). The cessation of ingression thus fits more closely with the
establishment of the tailbud about half way through body axis elongation. As
*Hoxb* and *d13* begin to be expressed just after
tailbud formation, they may contribute to conclusion of the ingression process, but this
early role also leaves room for other actions for such *Hox* genes,
including the regulation of cell proliferation in the tailbud mesoderm, described by
[19]. I think these
considerations are important to clarify that the regulation of ingression that they
uncover is distinct from the model they originally proposed in which this was directly
linked to “exhaustion” of PSM precursors and elongation arrest.

6) Related to this the authors suggest in the Discussion that posterior
*Hox* genes might act in place of retinoic acid as a mechanism for
arresting axis elongation in mouse. However, as ingression has ceased and continued
extension is due to the activity of internalized precursors it is difficult to see Hox
regulation of cell ingression as a molecular explanation for axis arrest.

7) The new data localize the region of *Hox* genes involved, but shows
that this does not contain a conserved sequence that correlates with repressive
activity. We are left with further speculation that this must be due to local protein
structure. Similarly, the authors may have identified a further part of the molecular
mechanism that regulates cell ingression, by rescuing *Hox* gene induced
delay by co-expression with the DN-RhoA construct. However, this simply shows that
de-stabilizing microtubules can force ingression and there is no molecular link made
here from Brachyury to regulation of RhoA activity. The authors have therefore made a
little progress on uncovering molecular explanations for the phenomena observed when
posterior *Hox* genes are mis-expressed.

8) Finally, the authors have attempted to determine whether *Sox2*
expression is present in epiblast cells that are delayed in their ingression through the
primitive streak. Unfortunately, they have been unable to improve the in situ
hybridization for *Sox2* transcripts (even the control embryo does not
reveal the normal caudal domain of *Sox2* expression, which extends into
the tailbud (e.g. see Uchikawa et al, 2011, Dev. Growth & Diff; [52]). The authors have
also been unable to detect *Sox2* protein using immunocytochemistry, and
it is not clear whether this reflects a problem with antibody batches or another
technical difficulty. The net result is that they do not provide any new data to address
the possibility that cells fail to ingress because having reduced Wnt signaling and
*T* expression they have lost mesodermal identity in favor of a neural
fate. Given that they are unable to reproduce the *Sox2* expression
pattern published by many other groups, they should be cautious in the interpretation of
their lower power in situ data, which leaves open the possibility that accumulating
non-ingressing primitive streak epiblast cells continue to express
*Sox2*.

[Editors' note: a further round of revisions was requested prior to acceptance,
as described below.]

Thank you for resubmitting your work entitled “*Hox* genes control
vertebrate body elongation by collinear Wnt repression” for further consideration
at *eLife*. Your revised article has been favorably evaluated by Janet
Rossant (Senior editor), a Reviewing editor, and two of the original reviewers. The
manuscript has been improved but there are some relatively minor but still important
issues that need to be addressed before acceptance, as outlined below:

1) The authors have responded with some discussion of the literature on when ingression
ceases, but this is not convincing. It is certainly debatable whether movement of cells
from the late chordo-neural-hinge (CNH) to the PSM is considered an ingression; the CNH
is the already internalized structure derived from the primitive streak. They should
clarify their interpretation.

2) The response to comment 8 is confused; expression of *Sox2* in
epiblast cells that fail to ingress does not automatically lead to their contribution to
the neural tube. Given the poor quality data (low magnification wholemount in situs
which do not recapitulate published caudal expression of *Sox2* even in
the controls), they should be cautious with the interpretation.

---

## [Author Response]

*1) One concern is the quantification of all the various phenomenon described in
this work. There are several parameters, which can easily escape control in this
context (mRNA stability, protein presence, electroporation success, copy number,
saturation etc.). This criticism is not particular to this work and a similar
question could be asked regarding the results by others in mice in mice, where the
strong effect was not explicitly related to a 'physiological' gain of
function (even though it may indeed be the case)*. *Vertebral formula
is very stable within the same species and thus must rely upon a robust mechanism.
How the robustness of this process relates to the quantities of factors is a key
issue (are these factors saturating?)*

We agree that the robustness of the data generated by our electroporation protocol is
striking. However, even though the technique does not allow us to precisely control the
number of plasmid copies per cell or the number of cells electroporated, we have well
established its reliability in the lab over the past 15 years. Successful
electroporations like microsurgical experiments require a thorough training, and Nicolas
Denans has been practicing these experiments on a regular basis for the past 6
years.

The quantitative analysis of the different phenotypes shown in the paper clearly
demonstrates that these experiments give highly consistent results with most embryos
overexpressing a given construct showing a similar phenotype which is specific for the
construct (see graphs in Figure 3, Figure 3—figure supplement 1, 6C, 9B).
Regarding mRNA stability, we overexpress only the coding sequences (without UTRs) and
use the same SV40 polyA signal for all of our constructs. Also in most cases,
*Hox* genes are expressed from an IRES-YFP vector and thus Hox
expression can be indirectly monitored by examining YFP fluorescence. All
*Hox* genes are expressed from the CAGGS promoter which is a strong
promoter and which is expected to generate expression levels higher than the endogenous
ones. Moreover, we clearly show in Figure 6
that the quantity of Hox protein overexpressed does not affect the phenotype observed.
So our conditions are saturating, abolishing any effect of gene dosage of the
overexpressed *Hox* genes and leaving the differences only due to the
nature of the *Hox* genes overexpressed. These observations are
consistent with the analysis of paralog knock-out experiments from Deneen Wellik (Wellik
et al, Science, 2003; McIntyre et al; Development, 2007)) showing that leaving only one
single wild type allele leads to a much milder phenotype than the deletion of an entire
paralog group. These experiments also confirm previous results published in Iimura and
Pourquie (Nature, 2006) showing that *Hoxb* genes expression driven by
promoters of different strength (CMV, TK and CAGGS) leads to similar ingression
phenotypes. Our data suggest that the quantitative effects on ingression result from the
progressive expression of more posterior genes. The fact that 8 of the 16 posterior
*Hox* genes from all posterior paralogs groups except
*Hox12* show an effect in the ingression, elongation and Wnt signaling
assays argue for an extreme redundancy of the system that could explain the
intraspecific robustness of the vertebral formula. We now discuss these ideas in more
detail in the revised version of the text.

*At the time of the start of decreasing Wnt signaling due to increasing Hox
amounts, a rather smooth gradient of T must be established*.

We have now analyzed by qPCR the expression of the Wnt targets T/Brachyury,
*Axin2* and *Fgf8* in the tail bud from the 10 to
25-somite stage (Figure8 C-E). These new data show that repression of Wnt targets
parallels activation of posterior *Hox* genes in vivo starting with a
smooth gradient until the 20-somite stage. A significant drop in expression is then
observed at the 25-somite stage, which corresponds to the time at which axis elongation
drops abruptly when *Hox13* genes are first expressed. This data has been
added to the text and is shown in Figure 8.

*At these time points, genetic conditions in mice would be expected to have an
effect perhaps stronger than what was reported by Capecchi for the* Hoxb13
*mutants?*

In null mutant mice embryos for *Wnt3a* or *T*, the first
7 to 9 somites form normally (Takada et al, 1994, Genes and Dev; Wilson et al,
Development, 1993), and the body axis is truncated posterior to this level. This
phenotype is much stronger than that reported for *Hoxb13* in which only
the tail part of the axis is affected. In the experiments performed by Deschamps and
Mallo where they overexpress *Hox13* paralog genes in the paraxial
mesoderm along the entire AP axis, these genetic conditions result in axis truncations,
occurring posterior to the thoracic level (Young et al, Dev Cell, 2009). Strikingly no
effect in the anterior regions is seen in these transgenic animals. The phenotype of
these animals resembles hypomorphic *Wnt3A* or *T* mutants
suggesting that the Wnt repression by posterior *Hox* genes is not an
on-off mechanism but rather a quantitative effect. This notion is further supported by
the different truncation levels reported for *Hoxa13*,
*Hoxb13,* and *Hoxc13* expressed from the same promoter
in mouse (Young et al, Dev Cell, 2009). These experiments are consistent with the notion
that Wnt and T play an important role in the control of axis elongation from the
cervical to the caudal level and that posterior *Hox* genes
quantitatively modulate Wnt activation in the paraxial mesoderm precursors of these
regions. We now discuss these points in the revised version of the manuscript.

*And if several* Hox *genes contribute to this phenomenon, then
how to prevent intra-species variations, in particular at the time-points when the
system would no longer be saturated?*

Our experiments reported here (Figure 6) and in
show that concentrations of Hox proteins exhibit a similar effect over a large dilution
range (Figure 6) suggesting that low levels of
Hox proteins are saturating. This property is expected to confer significant robustness
to the system in line with the limited intra-species variations observed.

*2) From this model, one can make rather strong predictions*,
*such as e.g. that the addition of yet another Hox cluster with posterior
genes should make the body shorter? Is this the case in the literature?*

It is true that zebrafish which has extra Hox clusters exhibits a shorter axis when
compared to mouse or chicken while snakes which contain less posterior
*Hox* genes expressed in the tailbud region (Di-Poi N. et al, Nature,
2010) make a much longer axis. However, whether this can be generalized will require
further studies. The Duboule lab has generated transgenic mice with a human HoxD cluster
or an extra mouse HoxD cluster (Spitz et al, Genes and Dev, 2001). Since the effect of
Hox proteins is saturating, duplicating an existing cluster is expected to act on
expression levels (rather than timing) of the Hox proteins and thus is not expected to
interfere with axis elongation. Accordingly, they did not observe any change in the
vertebral counts in these transgenic animals. They also have performed many duplications
and deletions within the HoxD cluster and shown that these manipulations can lead to
heterochronic expression of *HoxD* genes in the tail bud. However, they
do not report any significant variations in vertebral numbers in these cases either
(Tarchini et al, Dev Cell, 2006; Tschopp et al, PLOS Genet, 2009, Denis Duboule,
personal communication). Importantly, all these genetic manipulations have been
performed on the HoxD cluster and in our hands, neither *Hoxd12* nor
*Hoxd13* have an effect on elongation/Wnt, and the effect of
*Hoxd10* and *Hoxd11* is weaker than that of other
*Hox* genes thus providing a possible explanation for the lack of
significant effect of this heterochronic expression of posterior *HoxD*
genes. In contrast, heterochronic expression of other posterior *Hox*
genes such as *Hoxa13*, *Hoxb13* or
*Hoxc13* (but not *Hoxd13*) genes in the tail bud
precursors as described in Young et al (Dev Cell, 2010; and Jacqueline Deschamps,
personal communication) was shown to make the body shorter consistent with our
hypothesis. We now present our results in the light of these papers in the
Discussion.

*3) There are important differences in effects between the* Hoxa13
*and* Hoxd13 *genes. In many developmental contexts, these two
genes have very redundant functions and therefore, the difference reported here would
suggest that Hoxa13 achieves its function in the tailbud by affecting a pathway or by
using a peptide sequence that is not associated or existing in*
Hoxd13*? This is particularly unexpected, as* Hoxd13 *(at least
in mammals) seems to be expressed in the late tail bud more strongly than*
Hoxa13*. This should be discussed.*

In mouse, *a13* is the only *Hox13* required for embryonic
survival because its mutation blocks allantois growth is required for formation of the
placenta (Shaut et al, PLOS Genet, 2008). Also whereas transgenic mice overexpressing
*Hoxa13* from the cdx2 promoter exhibit axis truncation (Young et al,
Dev Cell, 2009), overexpressing *Hoxd13* from the same promoter has no
effect (Jacqueline Deschamps, personal communication). We now show that the effect of
posterior *Hox* genes on T repression is controlled by the N-terminal
domain of posterior Hox proteins (see text and new Figure 10). While we have not mapped precisely the sequences responsible for
the effects described in this paper, there are very significant differences in the
N-terminal regions of *Hoxa13* and *Hoxd13* that could
explain why in the control of ingression, of elongation and of Wnt signaling, these 2
proteins show different phenotypes (see alignment in Figure 12).Author response image 1.

*4) A major claim of the authors is that there is delayed ingression, but this is
not convincing. They report delayed expression of the Hox-Ires-Venus cassettes
following electroporation into the chick epiblast as measured by Venus protein
fluorescence. The authors interpret this effect such that Hox gene over-expression
can delay ingression of the cells receiving the construct. However, the Hox
constructs are driven by the ubiquitous Caggs promoter. Why is expression of the Hox
constructs not observed at the same time as that of the control construct? A simple
explanation might be that the RNA derived from these constructs is
post-transcriptionally controlled, either by immediate degradation or by
translational repression. This might assure position-adequate activity of
the* Hox *genes. RNA* in situ *hybridization
experiments testing for the time and position of transcription of the constructs are
missing. However, they are required at least for some of the constructs. In addition,
co-transfection of the control reporter construct with the Hox expression constructs
should reveal if indeed ingression is delayed by Hox over-expression. Furthermore,
RNAi of critical* Hox *genes (e.g.* Hoxa13*) would be a
means of testing if the proposed effect of* Hox *genes can be
opposed.*

We apologize if the description of our experiments was not clear enough leading the
reviewer to think that there is a delayed expression of the Hox-Ires-Venus cassettes
following electroporation into the chick epiblast as measured by Venus protein
fluorescence. We do not claim that expression is delayed but that ingression is delayed.
The two constructs (Hox-green and control-red) are electroporated one after the other
(see detailed description for consecutive electroporations below) and then we routinely
examine electroporated embryos 3 hours after electroporation to make sure that both
control and Hox constructs are expressed at the same time in the precursors of the
paraxial mesoderm in the anterior streak region. Then whereas control cells expressing
Cherry start to ingress in the posterior PSM, cells expressing the Hox-IRES-GFP
constructs remain in the epiblast of the anterior streak region ingressing only later.
Only a subset of posterior *Hox* genes shows a phenotype in these
experiments. This experimental paradigm was originally introduced in , see Figure 2, using grafts of epiblast expressing
different *Hox* genes. These experiments results in a gap separating the
position of the anterior boundary of red control cells from the position of the anterior
boundary of green Hox-expressing cells reflecting the fact that cells ingressing later
become located more posteriorly. This gap is proportional to the ingression delay
experienced by Hox-expressing cells. To quantitatively estimate the variation in the
ingression delay, we measure the ratio between the red and the green domains in the
paraxial mesoderm (Figure 3). We have tried to
improve the description of these experiments in the main text and in Material and
methods and provide a more detailed illustrative scheme (Figure 3).

Concerning RNAi, we have attempted to block the function of several posterior
*Hox* genes such as Hoxa13 by electroporating RNAi and in no case
could we observe a phenotype. This is not unexpected, given that RNAi only achieve
partial inhibition and that Hox proteins appear to be saturating even at low levels as
shown in Figure 6. This led us to turn to the
dominant-negative approach described in the paper.

*5) Critical missing data are the level of Brachyury expression along the body
axis. If the authors are right that increasing* Hox *gene expression
has a repressive effect on Brachyury expression resulting in decreasing mesoderm
formation and motility along the body axis, this should be reflected by decreasing
Brachyury transcription in the wild-type embryo as it elongates. Is this the case? Of
course this would mean measuring expression levels per mesoderm cell (e.g. FPKM or
qPCR values determined on FAC sorted cells) at different embryonic stages but at the
same axial position (e.g. close to the node), not just overall expression in the
embryo. The authors have all the tools at hand for doing the test.*

We agree with the referee and now provide this data. We have performed Quantitative-PCR
on microdissected tailbuds from 10 somite stage (ss), 15ss, 20ss, 25ss for
*T*, *Axin2* and *Fgf8* that clearly
show a down-regulation of the expression of these 3 genes between the 10-25ss stage.
This decrease parallels the slowing-down of axis elongation observed during those
stages. We now include these results in Figure 8
and discuss them in the main text.

*6) It is not clear if the data from chick can be extrapolated to vertebrates in
general. If the proposed mechanism were generally true, vertebrates with tails should
not exist. All* Hox *genes are already activated during trunk
development and the tail does not gain extra Hox activity grinding Brachyury
expression eventually down to a standstill. Thus in tailed animals a different
mechanism is more likely. The authors therefore should present their conclusions with
respect to the chick and possibly other vertebrates with tiny tails (humans, some
monkeys?).*

This is a very interesting point and an additional mechanism distinct from the one
reported here is possibly involved in tail growth. In chicken which lack a tail,
retinoic acid (RA) was shown to play an important role in the termination of axis
elongation and somitogenesis ([66]; Olivera-Martinez et al, 2011). This increase in RA level is accompanied by
Raldh2 reexpression in the tail bud and it triggers differentiation and death of the
paraxial mesoderm precursors. Remarkably, no such increase in RA levels is observed in
mouse embryos. Furthermore, a normal tail can form even in the absence of RA in the
*raldh2* mutant (13). Therefore, it could be that in mouse, termination of the axis and hence
tail formation, does not result from an active RA-dependent process but simply from
exhaustion of paraxial mesoderm due to the shrinking of the PSM resulting from posterior
*Hox* genes effect on elongation. Whether this RA-dependent mode of
axis termination can be more broadly generalized to species lacking a tail remains to be
explored. What causes this difference is currently unknown but it could be that in
chicken, specific genetic alterations of posterior *Hox* genes can
trigger the late expression of radlh2 in the tail bud thus leading to premature
termination of axis elongation. We now present these hypotheses in the Discussion of the
revised version.

*7) The accuracy of the experimental approaches is difficult to assess, in
particular, the accurate and reproducible targeting of the epiblast layer of the
primitive streak by electroporation is not documented. It seems likely that there is
some variability between embryos and that in some cases cells that are already
ingressing are targeted. Furthermore, it seems possible that control cells are
sometimes affected by Hox mis-expressing cells in the primitive streak, making this
assay difficult to interpret with confidence, particularly when only a handful of
embryos are analyzed for each gene. A data set demonstrating localized epiblast
targeting reproducibility would strengthen the claims of the authors that they are
assessing regulation of the ability of mis-expressing cells to ingress. It is also
concerning that the arrow indicating the most rostral red fluorescent label
in*
Figure 3
*is mis-placed caudally*.

We understand that this assay appears difficult and challenging and like microsurgical
experiments, these electroporations require a significant training to position the
electrode at the desired location appropriately. My lab has developed a significant
expertise in the electroporation of paraxial mesoderm precursors which we pioneered
(Dubrulle et al, Cell, 2001) and we have already published several papers based on this
technique (Iimura an Pourquie, Nature, 2006; Iimura et al, PNAS, 2007; Benazeraf et al,
Nature, 2010). Nicolas Denans has performed these electroporations on a regular basis
for the last 6 years now and he can obtain very reproducible results as shown on the
graphs in Figure 3, Figure 3—figure supplement 1, 6C, 9B. The electroporation
success is first monitored 3 hours after by examining the embryos under a fluorescent
stereomicroscope. Only embryos successfully electroporated in the PSM progenitors at the
exact same antero-posterior level and the same medio-lateral level relative to the
midline of the primitive streak are reincubated for about 20 more hours. About 10 to 20%
of the embryos fail to develop because of the 2 consecutive injections/electroporations.
The embryos that develop successfully are then examined for GFP expression and routinely
more than 90% show restricted YFP expression in the paraxial mesoderm. Such a high level
of success rate is obtained only after a long period of training. As can be seen on the
embryos shown in Figure 3, Figure 6, and Figure 9 or in
the movies, virtually no cells are seen outside of the paraxial mesoderm meaning that
the electroporation accurately targeted the anterior streak epiblast. Since expression
of the constructs is driven by the ubiquitous CAGGS promoter, even a slight inaccuracy
in positioning the electrode would result in expression in the neural tube or the
lateral plate. To better illustrate the accuracy of the electroporation performed, we
now present a movie showing expression of a control construct electroporated in the
anterior epiblast and showing the ingressing PSM cells (Video 3).

Our experiments have been carefully quantified as shown on the DotPlot graphs. As can be
seen in Figure 3 and Figure 7, when we overexpress a cherry control followed by a venus
control, cells always end up at the same AP position (with very minimal variations)
which shows that this assay is particularly robust. We electroporate one side of the PS
with one construct and the other side of the PS with the other construct which largely
prevents one cell to receive both plasmids. We agree that in some rare cases we could
electroporate some ingressing cells. These rare cells are escapers and are found located
much more anteriorly than the electroporated epiblast cells. We discard those rare cells
in our quantifications and Figure 3 is a good
example of it. We have now introduced a more thorough discussion of these technical
aspects in the text and in the Material and methods section.

*8) The time difference between*
“*successive*” *electroporations should be
clarified; if the first construct labels cells earlier then these will progress to a
more rostral position than cells electroporated slightly later, which could result in
a misleading conclusion that the second construct used delays ingression. It seems
the control construct is always electroporated first*; *can the
authors invert the order in which the constructs that induce a delay are
electroporated and obtain the same result?*

We now better describe our successive electroporation procedure (which we renamed
consecutive electroporation for clarity) in the Methods section. We first harvest and
culture the embryos on filter paper on an agar plate at room temperature (EC culture)
which “pauses” the development of the embryos. We then process to two
consecutive electroporations. We microinject the first construct and immediately
electroporate it and right after the first electroporation we proceed the same way for
the second construct. The time delay between the 2 electroporations is about 10 seconds.
Moreover as the experiments are carried out at room temperature, the embryo is paused
and cells do not ingress until the embryo is reincubated at 37 C. The fact that when we
overexpress 2 control constructs we find the cells at the same anterior position clearly
shows that the sequence of electroporation does not affect the timing of ingression
(Figures 3 and 7). In all cases we always
invert the order of electroporations in the same batch of embryos (but not the side of
electroporation for consistency) and do not observe any bias in the timing of
ingression). We now provide a better description of the consecutive electroporation
protocol in the text and Material and methods.

*9) There is no explanation at a molecular level for differences in the ability
of posterior* Hox *genes to delay ingression nor for the effect of
posterior dominance within the subset of posterior* Hox *genes that
appear to affect ingression. How does collinear onset of posterior* Hox
*genes lead to progressive loss of Wnt/FGF and T if this does not involve
protein accumulation? Identifying these molecular mechanisms is needed to take the
paper beyond the discovery of an interesting phenomenon.*

We agree that it is indeed a very interesting question. We designed chimeras of Hox
proteins to determine which domain contains the region responsible for the repressive
effect of posterior *Hox* genes on Brachyury using the cTprLuc assay as
readout. These experiments show that the N-ter region of *Hoxa13* contain
the repressive domain (see new Figure 10 in the
revised version of the manuscript). Sequence alignment of the N-ter region of
*Hoxa9*, *d10*, *c11* and
*a13* shows limited conservation at the amino acid level suggesting
that it is not a conserved amino acids domain but rather a structural domain that is
responsible for the repression activity of these proteins (see Figure 10—figure supplement 1). The situation we describe
is reminiscent of the recently described structural colinearity observed at the Hox
protein level by Richard Mann’s group (61). A more thorough structure function analysis of the protein
sequences for all these genes must be conducted but we feel like it falls beyond the
scope of the present paper.

*10) We are left at the end of the paper without a molecular explanation for
how* Hox *genes influence the ingression of cells. It is surprising
that a gene that leads to reduced Wnt and FGF signaling in primitive streak epiblast
does not affect cell adhesion and E-cadherin expression (data for which is not shown)
nor the balance of* Sox2/T *expression in this cell population which
would lead to cells adopting a neural over a mesodermal fate. Previous work has shown
that FGF signaling is upstream of E-cadherin downregulation, prompts EMT and
formation of T expressing mesoderm at the mouse primitive streak (Ciruna and Rossant,
2001, Dev. Cell), while more recent work has demonstrated that high Wnt signaling
promotes mesoderm differentiation from neuro-mesodermal (*Sox2/T
*co-expressing) progenitors and that reduced Wnt signaling in this context
leads to neural differentiation (*[67]*, Dev;*
[27]*,
PLOS Biology).*

Ciruna and Rossant nicely showed that FGF positively regulates *T* and
thus *FGFR1* mutation leads to T down-regulation consequently preventing
ingression, which correlates perfectly with our observations. Our data suggest that a
subset of posterior *Hox* genes trigger some down-regulation of Wnt/T and
FGF signaling which does not completely prevent ingression but delays it. We do not
claim that there is no overexpression of E-cadherin but only that we do not see ectopic
E-cadherin in cells overexpressing the Hox constructs. Our model, proposes that
*Hox* genes down-regulate *T* expression, delaying its
accumulation which is required for triggering ingression of epiblast cells by breaking
down the basement membrane and undergoing EMT (see Figure 4). However, when cells overexpressing posterior *Hox*
genes accumulate enough T protein they eventually ingress normally (like WT cells). As
stated above, we do not see cells overexpressing posterior *Hox* genes
activating *Sox2* (Figure 4)
nor entering massively the neural tube (see Figure 3, Figure 6, and Figure 9 or supplementary videos) arguing against a conversion of
epiblast cells to a neural fate by posterior *Hox* genes. Ingression of
cells during gastrulation requires disruption of a RhoA-dependent mechanism required for
stabilization of basal microtubules in epiblast cells (Nakaya Y et al, NCB, 2008). This
ultimately leads to loss of cell basement membrane interaction and breakdown of the
basement membrane allowing cell ingression. We now include experiments showing that
co-overexpression of a Dominant-negative-RhoA construct rescues the ingression phenotype
caused by *Hoxa13* overexpression (see Figure 4). This suggests that posterior *Hox* genes control
the timing of ingression by stabilizing the connection between microtubules and the ECM
rather than diverting epiblast cells to a neural fate. We now discuss these arguments in
more detail in the revised text.

*Other work, in chick, analyzing the expression of* Sox2 *and T
transcription throughout body axis elongation shows that strong T persists in the
tailbud long after the 25 somite stage, declining only just prior to cessation of
elongation at ∼ 44 somite after which T begins to decline and* Sox2
*is up regulated in tailbud cells, in a step that involves decreased FGF
signaling (*[52]*, PLOS Biology).*

*Indeed, Tenin et al, 2010, BMC Dev. Biol. show that elongation cessation takes
place when the presomitic mesoderm is still present at HH24 (rather than having been
used up by somites forming right to the tail tip), suggesting that the body axis does
not end due to a lack of ingressed cells. This observation contradicts the main
conclusion of this current paper that it is a lack of ingressed presomitic cells that
underlies elongation arrest*.

We do not claim that we have a complete downregulation of T but a slight decrease in
expression over time which is not incompatible with the papers cited by the reviewers.
The cited data has been obtained by ISH which is not a quantitative method. We now
provide a quantitative PCR analysis of *T* expression between 10 and 25
somites showing that T mRNA is decreasing during this period (Figure 8).

Our study emphasizes the role of *Hox* genes in regulating axis
elongation but does not address the arrest of somitogenesis per se even though our data
suggest that both are linked. This reviewer is right in saying that it is unlikely to
result from exhaustion of PSM precursors in the chicken embryo as there is still a small
amount of PSM remaining when axis elongation stops. This might however be the case in
mouse and we have amended our text accordingly. We have proposed that axis termination
is caused by exposure of the tail bud to retinoic acid produced by the somitic region,
which is made possible due to the shrinking of the PSM (Gomez et al, Nature, 2008). RA
has been shown to down-regulate FGF signaling and to induce differentiation and death of
the tail bud precursors leading to the arrest of elongation ([66], BMC Biol; Olivera Martinez et al, 2011, PLOS
Biol). In line with this hypothesis, we found that Cyp26A1 which is involved in RA
degradation is downregulated by *Hoxa13*, which was also observed by
Deschamps’s group in mouse (Young et al, Dev Cell, 2009). Whether the raise in
retinoids levels observed in the chicken embryo is caused by the shrinking of the PSM,
which brings the RA-producing somitic region next to the tail bud or by another
mechanism remains to be investigated. We now provide a more detailed description of
these arguments in the new version of the manuscript.

*11) The authors present only* Sox2 in situ *data (Figure
2–figure supplement 2) in whole mount embryos to support their claim that
there is not a cell fate change when* Hoxa13 *is mis-expressed.
Sections of such embryos in which the* Hoxa13 *GFP construct is
visible with the* Sox2 in situ *signal on a cell by cell basis is
required for convincing evidence on this point. A further possibility is
that* Hoxa13 *expressing cells simply down regulate T but
remain* Sox2 *positive and so adopt a neural fate and therefore do not
ingress. Again comparing control and* Hoxa13 *GFP positive cells for
levels of T and* Sox2 *on a cell-by-cell basis may provide evidence
for the decline in T and maintenance* Sox2*.*

We performed sections on embryos electroporated with control or *Hoxa13*
vectors and processed them for ISH with a *Sox2* probe but the signal was
too weak to be convincingly detected in sections. We also tried to perform
immuno-histochemistry for *Sox2* on tailbud sections but failed to find
an antibody that faithfully labels the *Sox2* population in the CNH. We
tried the antibody published in ([52], PLOS Biology) but in our hands it only gives a strong background
even if we dilute the antibody to 1/2000 (in the paper they use it at 1/200). It is a
polyclonal antibody so it might be related to batches issues (we tried 2 batches).
Importantly, the descendants of the epiblast cells overexpressing posterior
*Hox* genes are always found in the paraxial mesoderm and rarely in
the neural tube (see Figure 3, Figure 6, and Figure 9 or supplementary videos). This further indicates that Hox overexpressing
cells do not convert to a *Sox2* positive neural fate. We have added this
important argument to the text of the revised version.

*12) It is not clear why the authors provide* in situ *images
for* T *and* Fgf8 *at HH14 (*Figure 5*) using an intronic
probe; other genes are not assessed in this way and no comment or explanation is
provided in the text or figure legend. It is not clear how this data compares with
published work showing strong and extensive T transcripts until the end of axis
elongation. The authors need to address this difference with previous work, explain
the significance of loss of active T transcription and consider how T transcripts
continue to be regulated by FGF in the tailbud at these late stages (*[52]*) if the gene is not actively transcribed. One logic may
be that the effects of endogenous posterior* Hox *genes are very small
and slow acting, however, the authors appear to argue that there is a step change as
the tailbud forms and the image presented in*
Figure 5
*implies that T transcripts are lost in the tailbud at this time.*

*T* and *Fgf8* are highly expressed genes and the exonic
probes for these genes saturate extremely rapidly, even when we develop the embryos at 4
degrees, which makes it extremely difficult to quantify a slight change in gene
expression. This is why we decided to use intronic probes which showed a downregulation
of the signal which parallels the slowing down of axis elongation. We now also provide a
quantitative PCR analysis for these genes from microdissected tail buds in Figure 8 that confirms our ISH data. This new data
confirms small progressive downregulation of T with a significant drop at the 25 somite
stage when the first *Hox13* genes are activated.

*[Editors' note: further revisions were requested prior to acceptance, as
described below*.*]*

*[…] Overall, this paper presents a striking phenomenon in the regulation
of cell ingression through the primitive streak by posterior* Hox
*genes and some very interesting data that explain to some extent the
molecular mechanisms at play here. The authors still need to place the work in an
appropriate context both in terms of the literature and the significance of this
mechanism during body axis elongation; it may be that collinear activation of
posterior* Hox *genes regulates tailbud morphogenesis half way through
axis elongation, but does not directly regulate PSM length.*

*1) The role of RhoA is not sufficiently well investigated and documented for
giving it a prominent role in the Abstract. If the authors find this point so
important they should provide some more experimental support. Alternatively they may
want to rewrite the Abstract*.

We have deleted the sentence referring to RhoA in the Abstract.

*2) Last line of the Abstract (end of the Introduction and elsewhere,):*
“*this mechanism leads to progressive reduction of PSM size associated
to the arrest of somitogenesis*” *does not follow logically.
What does this mean if it is clear that, at least in the chick, body axis elongation
ceases when the PSM is still present and there is room for more somites to be
generated? How is ingression then associated with arrest of somitogenesis?
Somitogenesis arrest is more likely linked to loss of Notch pathway oscillations in
the remaining PSM (*[66]*)*.

We agree that the formulation was unclear and we have changed the last sentence of the
Abstract into: “Due to the continuation of somite formation, this mechanism leads
to the progressive reduction of PSM size. This ultimately brings the retinoic acid
(RA)-producing segmented region in close vicinity to the tail bud, potentially
accounting for the termination of segmentation and axis elongation”. We now
expand on these notions in the Introduction and Discussion as discussed in point 3
below.

*3) In the Introduction the authors should mention the evidence that explanted
chick tailbuds (i.e. without somites) are a demonstrated source of RA (*[66]*). This
is an important finding, which bears on their hypothesis that shortening of the PSM
is a major step in that brings retinoic acid to tailbud and so curtails body axis
elongation. It may be that both mechanisms operate and I think this should be made
clear upfront, rather than appearing later in the Discussion*.

We have added these findings and discuss them more extensively in the Introduction. The
following text has been added to the same part of the text:.

“In chicken and fish embryos, the arrest of axis elongation has been linked to
the inhibition of FGF and Wnt signalling in the tail-bud which leads to the
down-regulation of the transcription factor *T/Brachyury* and of the
Retinoic Acid (RA)-degrading enzyme *Cyp26A1* (77; 42; 66;
52). […]
The shrinking of the PSM which brings the segmented region producing RA in the vicinity
of the tail bud might also contribute to the raise in RA levels in the tail bud and
possibly to the late *Raldh2* activation in the tail bud.”

We have also added the following text to the revised Discussion:

“Furthermore, the inhibition of FGF and Wnt signaling which are required for the
segmentation clock oscillations […] is also responsible for
*Raldh2* activation in the late tail bud remains to be
explored.”

*4) In the Introduction and elsewhere it is confusing to state that*
“Hoxb *genes are only expressed in anterior regions of the embryo
precluding their playing a role in the control of axis
extension*”*, when* Hoxb13 *is known to regulate
axis elongation. They need to be more precise,* Hoxb1-9*.*

We agree and have modified the text accordingly.

*5) It would be informative to explain in the paper that the 25 somite stage is
∼when the tailbud forms and that this is when when cell ingression through the
primitive streak ceases—refer to work from Susan Mackem's lab in chick
showing cessation at HH stage16 (∼ 24-28 somites,*
[37]*,
and see*
[9]*, and
also*
[73]*, in the mouse). It seems likely that the mechanism
identified by the authors is most functionally significant at this mid-way stage;
from then on the PSM is added to from an*
“*internalized*” *set of precursor cells. McGrew
et al 2009 and*
[52]
*identify these cell populations by lineage analysis in the chick tail bud as the
NMps (Sox/bra expressing cells) and also a transit amplifying cell population of
mesodermal progenitors. I think the authors should explain this in the Discussion
(and in the Introduction). As the paper currently reads it might be thought that
ingression through the primitive streak continues until the end of axis elongation,
when in fact in both chick and mouse embryos this ceases at somite 30 of
∼50-53 (chick, HH16) and 33 of 65 (mouse, E10.5). The cessation of ingression
thus fits more closely with the establishment of the tailbud about half way through
body axis elongation. As* Hoxb *and* d13 *begin to be
expressed just after tailbud formation, they may contribute to conclusion of the
ingression process, but this early role also leaves room for other actions for
such* Hox *genes, including the regulation of cell proliferation in
the tailbud mesoderm, described by*
[19]*.
I think these considerations are important to clarify that the regulation of
ingression that they uncover is distinct from the model they originally proposed in
which this was directly linked to*
“*exhaustion*” *of PSM precursors and elongation
arrest.*

Our model argues that posterior *Hox* genes regulate axis elongation by
controlling cell ingression in the PSM (and also by regulating cell motility,
independently of ingression) and thus makes the implicit assumption that this ingression
process continues during axis elongation. Whether this is the case or not as raised by
this comment is an excellent point. However, we do not agree with the assertion that in
the chicken embryo, cell ingression ceases at stage 16HH (25-somite stage). While this
has been published in the Knezevic paper in 98, since then Ohta et al (Gen
Yamada’s lab) have published in 2007 in Development that ingression continues at
the level of the Ventral Ectodermal Ridge (the remnant of the primitive streak which
forms around the 25-somite stage) at least up to stage 20 HH (40-43 somites). More
recent data from Kate Storey’s lab (Olivera-Martinez et al, PLOS Biol, 2012) also
show ingression from the late chordo-neural hinge as late as stages 21-22HH (>45
somites). Overall, very little is known about the movement of cells in the tail-bud
after the 25-somite stage in the chicken embryo.

We now provide an extended discussion of the literature concerning ingression and
paraxial mesoderm precursors. The text below has been added to the Discussion of the
revised version:

“In the chicken embryo, PM precursors originate initially from the lateral
epiblast which migrate toward the midline during formation of the primitive streak (
[59]; [31]). […] Whether
*Hox13* genes might regulate the late ingression of PM at the level of
the VER as shown by Ohta or at the level of the CNH as proposed by Olivera-Martinez
remains to be established. In both cases, however, the PSM is expected to shrink in
response to *Hox13* genes.”

*6) Related to this the authors suggest in the discussion that posterior*
Hox *genes might act in place of retinoic acid as a mechanism for arresting axis
elongation in mouse. However, as ingression has ceased and continued extension is due
to the activity of internalized precursors it is difficult to see Hox regulation of
cell ingression as a molecular explanation for axis arrest*.

In mouse [73] initially
reported an arrest of ingression when the posterior neuropore closes at the 30-somite
stage, but subsequently Wilson and Cambray (2002) provided evidence for mesoderm
ingression after neuropore closure. In addition, we also show that posterior
*Hox* genes act on cell motility in the PSM, which by itself might be
sufficient to explain some slowing down of elongation.

To clarify this point, we have added the following text to the revised Discussion:

“In mouse embryos, [73] initially reported an arrest of ingression when the posterior neuropore
closes at the 30-somite stage (73), but Wilson and Cambray (2002) subsequently provided evidence for
continued ingression of cells in the PM after this stage (Cambray and Wilson, 2002).
Thus, in mouse embryos, termination of axis elongation could simply result from
exhaustion of PM progenitors caused by the slowing of axis elongation triggered by
posterior *Hox* genes acting on cell ingression and motility.”

*7) The new data localize the region of* Hox *genes involved, but
shows that this does not contain a conserved sequence that correlates with repressive
activity. We are left with further speculation that this must be due to local protein
structure. Similarly, the authors may have identified a further part of the molecular
mechanism that regulates cell ingression, by rescuing* Hox *gene
induced delay by co-expression with the DN-RhoA construct. However, this simply shows
that de-stabilizing microtubules can force ingression and there is no molecular link
made here from Brachyury to regulation of RhoA activity. The authors have therefore
made a little progress on uncovering molecular explanations for the phenomena
observed when posterior* Hox *genes are mis-expressed.*

We feel that there is already a very significant amount of data in the paper
characterizing the properties of a set of *Hox* genes on axis elongation.
While it will be interesting to perform structure function analysis, we feel this is
beyond the scope of the paper. Moreover, *Hox* genes have been first
cloned in the 80’s and to date, very little is still understood of their function
as transcription factors.

*8) Finally, the authors have attempted to determine whether* Sox2
*expression is present in epiblast cells that are delayed in their ingression
through the primitive streak. Unfortunately, they have been unable to improve
the* in situ *hybridization for* Sox2 *transcripts
(even the control embryo does not reveal the normal caudal domain of* Sox2
*expression, which extends into the tailbud (e.g. see Uchikawa et al, 2011,
Dev. Growth & Diff;*
[52]*). The authors have also been unable to detect*
Sox2 *protein using immunocytochemistry, and it is not clear whether this
reflects a problem with antibody batches or another technical difficulty. The net
result is that they do not provide any new data to address the possibility that cells
fail to ingress because having reduced Wnt signalling and* T
*expression they have lost mesodermal identity in favor of a neural fate.
Given that they are unable to reproduce the* Sox2 *expression pattern
published by many other groups, they should be cautious in the interpretation of
their lower power* in situ *data, which leaves open the possibility
that accumulating non-ingressing primitive streak epiblast cells continue to
express* Sox2*.*

If the cells overexpressing posterior *Hox* genes were retained in the
epiblast because of a switch to a neural identity, then one would expect to see the
descendents of these cells in the neural tube. This is absolutely not the case as the
overexpressing cells massively enter the PSM. Together with our low magnification
*Sox2* in situs that do not show upregulation of *Sox2*
in overexpressing cells, we believe that this strongly argue against
*Hox* genes maintaining cells in the epiblast by converting them to a
neural fate. Furthermore as shown on the movies, we clearly see ingression of Hox
expressing cells in the PSM.

*[Editors' note: a further round of revisions was requested prior to
acceptance, as described below*.*]*

*The manuscript has been improved but there are some relatively minor but still
important issues that need to be addressed before acceptance, as outlined
below*:

*1) The authors have responded with some discussion of the literature on when
ingression ceases, but this is not convincing. It is certainly debatable whether
movement of cells from the late chordo-neural-hinge (CNH) to the PSM is considered an
ingression; the CNH is the already internalized structure derived from the primitive
streak. They should clarify their interpretation*.

When it comes to ingression passed the primitive streak stage, the situation is
extremely unclear. We have nevertheless attempted to clarify the situation as suggested
by modifying the original conclusion paragraph as shown below:

“There is also some lineage continuity at the level of the PM precursors of the
Node/primitive streak border […]. Overall, very little is known about the
movements of cells in the tail-bud after the 25-somite stage in chicken and mouse
embryos.”

*2) The response to comment 8 is confused; expression of* Sox2 *in
epiblast cells that fail to ingress does not automatically lead to their contribution
to the neural tube. Given the poor quality data (low magnification wholemount in
situs which do not recapitulate published caudal expression of* Sox2
*even in the controls), they should be cautious with the
interpretation.*

We have now modified the text in conclusion as shown below to add to more arguments
against the fact that posterior *Hox* genes induce cells towards a
*Sox2*-positive neural plate, hence preventing their ingression,
namely the fact that *Sox2* is not detected in the microarray analysis of
*Hoxa13* overexpressing cells and the fact that Hox-ingressing cells
are clearly observed as shown in Figure 4. As
suggested, we have nevertheless added a sentence of caution at the end of the
paragraph:

“Thus the Wnt repression experienced by epiblast cells in response to posterior
*Hox* genes overexpression could induce these cells toward a neural
fate hence preventing them to ingress. However, *Hoxa13* overexpression
does not lead to up-regulation of the neural marker *Sox2* in
electroporated cells as detected by in situ hybridization and in our microarray
analysis. Furthermore, electroporated cells are seen to enter the PM and do not enter
the neural tube (Figure 4 and supplementary
movies). This therefore suggests that posterior *Hox* genes are unlikely
to control cell ingression by promoting acquisition of a neural fate in epiblast cells.
While we cannot completely rule out that a subpopulation of these cells remains in the
tail-bud as *Sox2*-positive cells, it is unlikely that this contributes
to the dramatic axis elongation slow-down observed after posterior *Hox*
genes overexpression.”